# Diaphragm-based carbon monoxide electrolyzers for multicarbon production under alkaline conditions

Wanyu Deng [1,3], Siyang Xing[1,3], Guilherme Warwick Parker Maia [1], Zhaoxi Wang [1], Bradie S. Crandall[2] & Feng Jiao [1] ✉

Transforming waste carbon into valuable fuels and chemicals is a key step toward sustainable manufacturing. One promising approach is the electrochemical conversion of carbon monoxide (CO), a product of $CO_2$ recycling, into energy-rich multicarbon ($C_{2+}$) compounds. However, current CO electrolyzers rely on anion exchange membranes (AEMs) that degrade over time when exposed to organic intermediates, limiting their practical use. Here we show that low-cost diaphragm materials, such as Zirfon, can serve as robust alternatives to AEMs in alkaline CO electrolysis. We evaluate a range of diaphragms and identify candidates that match or exceed the performance of commercial AEMs across a wide range of operating conditions (50 to 400 mA cm$^{-2}$). At 60 °C, Zirfon-based cells maintain 45% Faradaic efficiencies for acetate over 250 hours, while state-of-the-art AEMs fail within 150 hours. Moreover, a 100 cm$^2$ Zirfon cell operates stably for 700 hours at 200 mA cm$^{-2}$. These findings demonstrate that diaphragms offer a scalable and durable pathway for CO electrolysis, helping reduce system costs and enhance compatibility with renewable energy inputs.

Carbon monoxide (CO) electrolysis is a critical technology for carbon utilization, enabling the transition toward a carbon circular economy by converting CO into valuable $C_{2+}$ products[1-4]. Currently, CO electrolyzers often rely on AEMs to facilitate ion transport and separate reaction products[5-8]. However, commercial AEMs are not specifically designed to withstand the organic intermediates, such as alcohols and aldehydes, generated during $CO_2$ and CO electrolysis[9]. This unique environment significantly differs from more established systems like water electrolyzers and hydrogen fuel cells, where intermediates are typically less aggressive[9,10]. Consequently, the incompatibility of AEMs with organic intermediates dramatically shortens membrane lifetime, severely limiting the durability of current-generation CO electrolyzers[9,11-13]. Beyond durability concerns, AEMs are still relatively expensive, representing another barrier to scalable commercialization[14-16]. Therefore, identifying and developing low-cost, chemically resilient alternatives to conventional AEMs is essential

to achieve not only enhanced device stability and efficiency but also significantly low overall system costs, improving the commercial viability of CO electrolysis technology[15,17].

Diaphragms are widely used in alkaline water electrolysis (AWE)[18-23], a commercially mature technology for hydrogen production. In AWE, the diaphragm serves as a porous separator that physically divides the anode and cathode compartments while allowing ionic conductivity through the migration of hydroxide ions (OH$^-$) in the alkaline electrolyte[24-26]. This diaphragm-based cell configuration enables the electrochemical splitting of water into hydrogen and oxygen while minimizing gas crossover[24,27,28]. Unlike polymer-based membranes, diaphragms are typically made of inexpensive inorganic materials, such as zirconia-reinforced asbestos or polysulfone composites, which exhibit high chemical stability under alkaline conditions[24,29,30]. Their low cost is a major advantage, contributing to a lower capital expenditure for AWE systems compared to competitive

[1]Center for Carbon Management, Department of Energy, Environmental, and Chemical Engineering, McKelvey School of Engineering, Washington University in St. Louis, St. Louis, MO, USA. [2]Lectrolyst, Wilmington, DE, USA. [3]These authors contributed equally: Wanyu Deng, Siyang Xing. ✉e-mail: jiaof@wustl.edu

technologies[31–33]. Since CO electrolysis is also typically performed under alkaline conditions and shares similar ion transport requirements[34], diaphragms present a promising, low-cost alternative to AEMs[22], which are prone to degradation from reactive organic intermediates.

Here, we report a diaphragm-based CO electrolyzer design (Fig. 1), where a gas-diffusion electrode with Cu nanoparticles is used as cathode and an anode is based on NiFeO$_x$ supported on Ni foam. A variety of diaphragm materials were evaluated, including commercial Zirfon materials, polyethersulfone (PES), Nylon, polyvinylidene fluoride (PVDF), and fiberglass. All candidates except fiberglass showed a high CO electroreduction (COR) performance, with faradaic efficiencies (FE) for C$_{2+}$ products comparable to those achieved using state-of-the-art AEMs. The diaphragm-based electrolyzers also operated effectively across a broad current density range (50-400 mA cm$^{-2}$), maintaining high C$_{2+}$ product selectivity (~80%). At 60 °C, the 5 cm2 Zirfon-based cell exhibited minimal performance degradation over 250 h, sustaining an acetate FE of 45%, while the PiperION based cell failed within 150 h, with hydrogen FE increasing to 60%, indicating loss of selectivity. Furthermore, we successfully scaled the Zirfon-based CO electrolyzer from a 5 cm$^2$ lab-scale cell to a 100 cm$^2$ device, achieving 700 h of stable operation with a 50% acetate FE at 200 mA cm$^{-2}$. These findings highlight the potential of diaphragm materials, as cost-effective and durable alternatives to AEMs for CO electrolysis.

## Results and discussion
### Diaphragm materials for CO electrolysis

All diaphragms and AEMs were evaluated using a zero-gap CO electrolyzer (Fig. 1 and Supplementary Fig. 1a) with a 5 cm$^2$ active electrode area. This configuration was chosen for its low ohmic resistance and high reaction rate, while providing a scalable and commercially relevant platform[35–38]. Cu nanoparticles (40-60 nm in diameter, Sigma-Aldrich) supported on a carbon gas diffusion layer served as the cathode for COR, while NiFeO$_x$/Ni foam was used as the anode for oxygen evolution reaction. Diaphragms were used in place of AEMs to separate the cathode and anode. The porous structure of the diaphragms is infiltrated with KOH electrolyte (1 M or above), enabling facile transport of key ionic species (K$^+$ and OH$^-$) within the electrolyzer (Fig. 1). This setup provides a highly alkaline environment at the cathode, which suppresses the competing hydrogen evolution reaction and improves the selectivity of COR.

We first evaluated the product selectivity for COR using five diaphragm materials, Zirfon, PES, Nylon, PVDF, and fiberglass. Their performances were compared with that of commonly used AEMs in CO electrolysis at a current density of 200 mA cm$^{-2}$ (Fig. 2a). Except for fiberglass, nearly all tested diaphragms exhibited a low hydrogen FE and a comparable COR selectivity to those of AEMs (Fig. 2a). Post-reaction scanning electron microscopy (SEM) images (supplementary Fig. 2) taken after 120 h of continuous operation at 200 mA cm$^{-2}$ revealed no visible structural changes in Zirfon 500 + , PES, and Nylon diaphragms. In contrast, fiberglass and PVDF showed clear signs of structural and/or color changes (Supplementary Fig. 3), indicating material degradation under alkaline electrolysis conditions[39,40]. Additionally, similar X-ray photoelectron spectroscopy (XPS) spectra obtained pre- and post-reaction for Zirfon 500 + , PES, and Nylon diaphragms (Supplementary Fig. 4) suggest their structural stability during the CO electrolysis. All these three diaphragm materials were also evaluated under a wide range of current densities (50-400 mA cm$^{-2}$), where they exhibited C$_{2+}$ FEs exceeding 80% (Fig. 2b and Supplementary Fig. 5). The product distributions observed in diaphragm systems closely match those

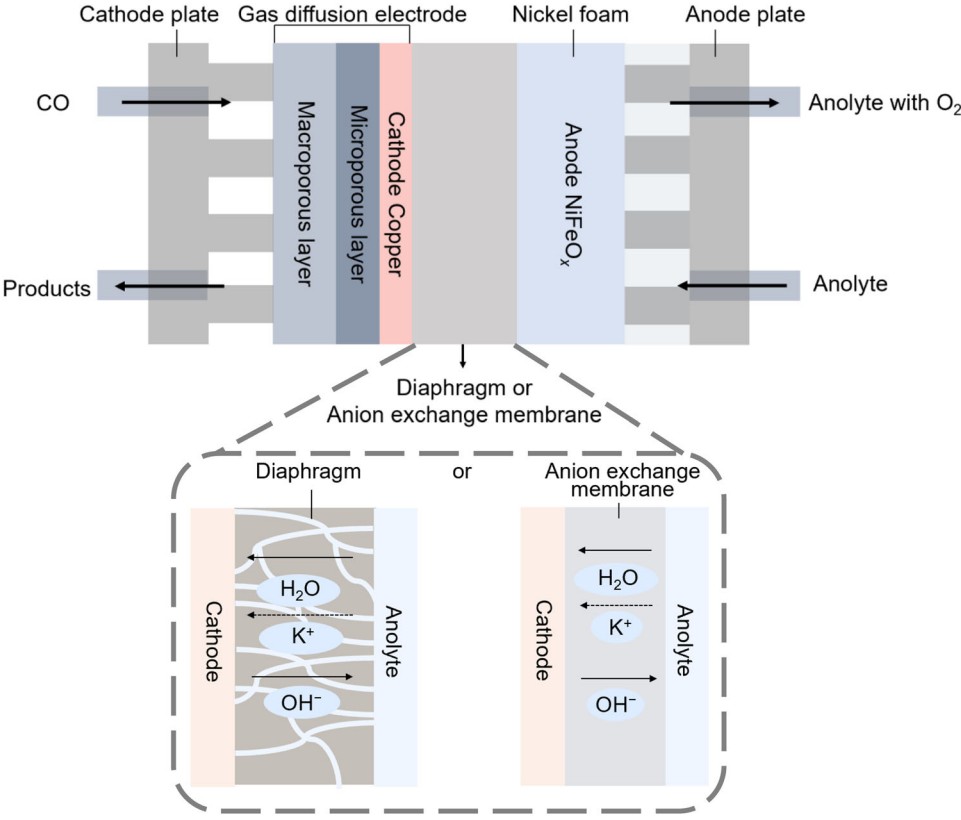

**Fig. 1 | Schematic diagram of zero-gap CO electrolyzer based on a diaphragm or an anion exchange membrane with KOH electrolyte.** Cu served as the cathode, and NiFeO$_x$/Ni foam was used as the anode. Diaphragms with defined pore structures were infiltrated with KOH electrolyte, enabling the transport of ionic species (such as K$^+$ and OH$^-$). Anion exchange membranes (AEMs) enabled anion-selective transport between the anode and cathode.

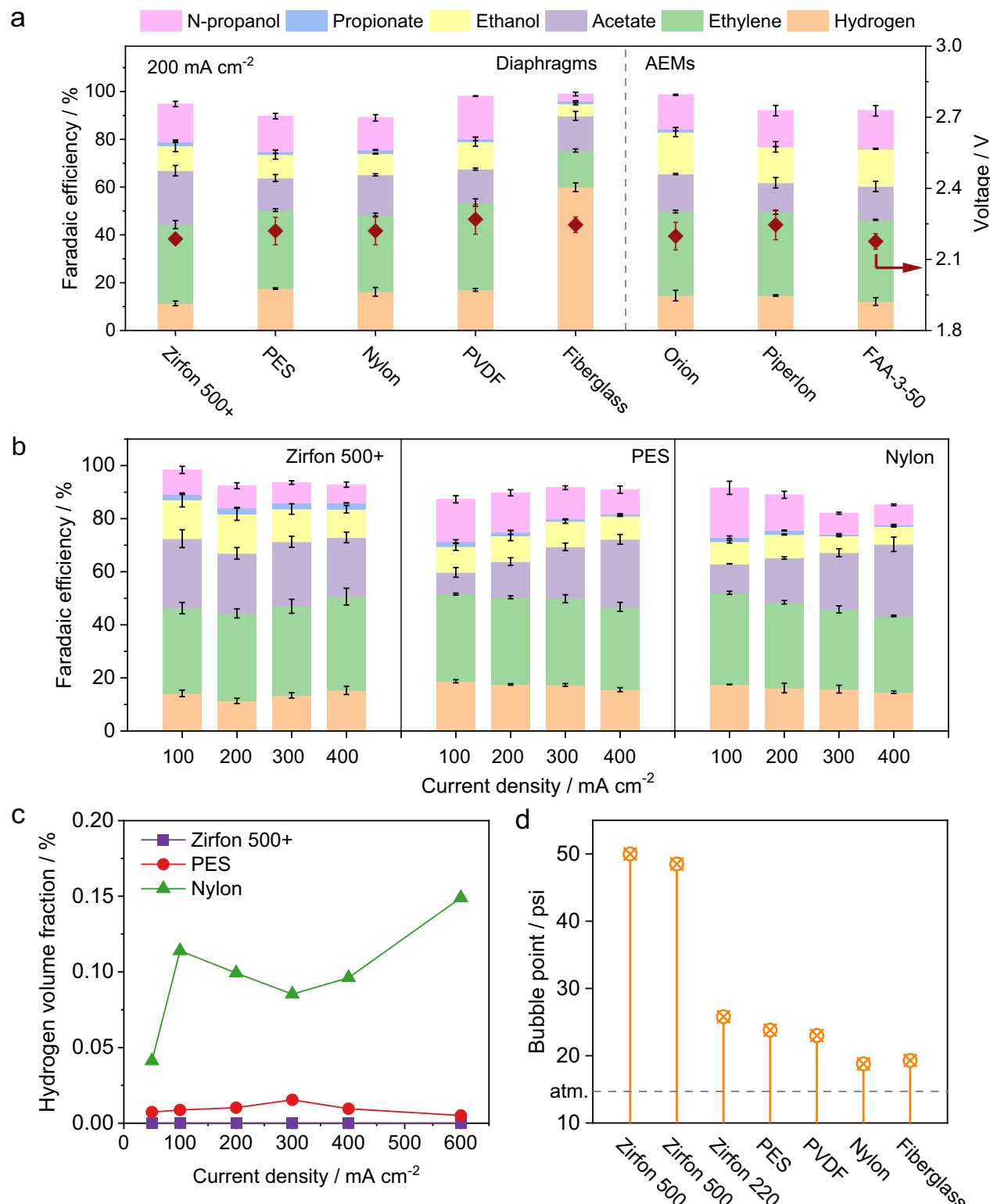

**Fig. 2 | CO electrolyzer performance with different diaphragms and AEMs.**
**a** Faradaic efficiency for all detectable products and corresponding cell voltages measured using different diaphragms and AEMs with a constant current density of 200 mA cm$^{-2}$. **b** Comparison of CO electroreduction performance with different diaphragms among different current density. **c** Volume percentage of hydrogen crossover from the anode to cathode as a function of current density for different diaphragms. There is no detectable hydrogen crossover in the case of Zirfon 500 + .

**d** Bubble point comparison of different diaphragms. The 5 cm$^2$ zero-gap diaphragm-based CO electrolyzer was operated with a 1 M KOH electrolyte at 3 mL min$^{-1}$, a 40–60 nm Cu nanoparticle cathode, a NiFeO$_x$/Ni foam anode, with CO fed at a rate of 30 sccm. Error bars in (**a**) and (**b**) represent the standard deviation from three independent measurements. Source data are provided as a Source Data file.

with AEMs (Supplementary Fig. 6), highlighting the viability of these diaphragms as potential low-cost alternatives to AEMs for scalable and practical CO electrolysis applications. For PES diaphragms, we examined a series of materials with a pore size ranging from 0.1 to 1.2 μm exhibited similar FEs and cell voltages (Supplementary Fig. 7).

Gas crossover is a critical issue in diaphragm-based systems, especially at higher current densities and during prolonged operation[41]. Therefore, we investigated gas crossover in both diaphragms and AEMs under varying current densities by analyzing the gas effluent from the anode side after 24 h experiments (Fig. 2c). No detectable gas (oxygen or hydrogen) crossover was observed for Zirfon 500+ at any tested current density (Supplementary Figs. 8, 9). In contrast, all other diaphragms and AEMs exhibited various degrees of hydrogen crossover (no oxygen detected). Quantitative analysis of the hydrogen concentration in the anode gas stream revealed that, the volume fraction of hydrogen for all AEMs and diaphragms remained less than 0.09%, below the lower explosion limit of 4%[42]. To further determine whether crossover hydrogen is oxidized at anode, a separate hydrogen oxidation reaction (HOR) cell with an identical zero-gap configuration was connected downstream of the CO electrolyzer (Supplementary Fig. 10a). A full cell potential of 1.48 V that is slightly lower than the open-circuit voltage (~1.52 V) of the CO electrolyzer was applied to the HOR cell (Supplementary Fig. 10b), and the resulting current response was used to assess the extent of HOR[43]. All tested diaphragms exhibited current responses below −0.05 mA cm$^{-2}$ (Supplementary Fig. 10c), indicating that internal HOR is not a significant side reaction under the test conditions.

Additionally, we also conducted bubble point measurements for each diaphragm material to further evaluate potential gases crossover in diaphragm systems (Fig. 2d). Zirfon 500+ and Zirfon 500 diaphragms exhibited relatively high bubble points (~50 psi), whereas all other diaphragms showed values below 30 psi (Fig. 2d and Supplementary Fig. 11), likely due to their reduced thickness (~130 μm compared to 500 μm for Zirfon 500+ and Zirfon 500). These results also indicate that the Zirfon-based diaphragms offer superior gas barrier properties compared to other diaphragm materials. Therefore, subsequent investigations focus primarily on the performance of Zirfon-based diaphragms and their comparison with commercially available AEMs.

## Performance of Zirfon diaphragms

Given the promising performance of Zirfon 500+ as a diaphragm material for CO electrolysis compared to other materials, we conducted a detailed evaluation of three commercially available variants: Zirfon 220, Zirfon 500, and Zirfon 500+ (Supplementary Table S1 and Supplementary S12). Both Zirfon 500 and Zirfon 500+ exhibited relatively high bubble points (~50 psi; Supplementary Fig. 11), attributable to their greater thickness (500 μm, Supplementary Fig. 12). However, the increased thickness may also result in a higher ionic resistance, and thus a lower energy efficiency. Therefore, we measured the area-specific resistance of the Zirfon diaphragms in 1 M KOH at both 25 and 60 °C (Fig. 3a and Supplementary Fig. 13). Zirfon 500+ displayed a lower resistance than that of Zirfon 500, likely due to its higher porosity (60%, Supplementary Table S1). Although the Zirfon 500+ diaphragm still has a higher area-specific resistance than that of PiperION membrane (Fig. 3a), the full cell voltage of the Zirfon 500+ based cell is slightly lower than that with the PiperION based cell at all current densities (Fig. 3b). To elucidate the origin of this discrepancy, we conducted voltage breakdown analyses[44] using a five-electrode configuration for both systems in 1 M KOH (Supplementary Figs. 14–17). The analysis revealed that the membrane ohmic resistance accounts for only a small fraction of the total cell voltage, with over 95% of the voltage drop arising from the cathodic and anodic non-ohmic and thermodynamic components (Supplementary Figs. 16–19). In terms of COR performance, all Zirfon diaphragms demonstrated comparable

$C_{2+}$ selectivity and operating voltages relative to the membrane-based system (Supplementary Fig. 20). In addition, no significant difference in product purity was observed between the two systems (Supplementary Fig. 21).

To assess the durability of Zirfon 500+ in CO electrolysis, we performed experiments using a 5 cm$^2$ CO electrolyzer at 200 mA cm$^{-2}$. The electrolyzer was operated at room temperature without any external cooling or heating. The Zirfon 500+ based cell exhibited stable performance over a continuous 700 h operation, maintaining a 30% ethylene FE and 50% acetate FE (Fig. 3c). In contrast, the PiperION based cell lost its $C_{2+}$ product selectivity in less than 150 h (Supplementary Fig. 22), showing a substantially lower durability than the Zirfon 500+ configuration. SEM and XPS analysis were conducted to probe structural changes in electrolyzer components pre- and post-stability test. SEM images revealed no noticeable morphological changes for either Zirfon 500+ or PiperION after electrolysis (Supplementary Fig. 23). However, PiperION membrane exhibited an obvious color change from transparent to orange, indicating potential chemical degradation (Supplementary Fig. 24). XPS analysis of post-reaction cathode surface shows no signals corresponding to Ni or Fe, confirming no cathode contamination from Ni or Fe leaching at the anode (Supplementary Fig. 25). Additionally, XPS and FTIR revealed the disappearance of the N 1s peak initially present in the pristine PiperION membrane (Supplementary Figs. 26, 27), suggesting the degradation of PiperION during COR. Conversely, Zirfon 500+ exhibited no notable XPS spectral changes, confirming its high chemical stability under identical conditions (Supplementary Fig. 26).

## Zirfon-based CO electrolyzer operated at elevated temperatures

In commercial AWE, Zirfon diaphragms typically operate at elevated temperatures (60-80°C) and high KOH concentrations (4-6 M). Thus, we further assessed Zirfon 500+ performance under these conditions for CO electrolysis. The full-cell potential decreases with rising temperatures up to 80 °C (Fig. 4a), likely due to enhanced ionic conductivity and reduced internal resistance[45,46]. Interestingly, although the total $C_{2+}$ FE remains relatively similar at higher temperatures, ethylene selectivity notably increases at all current densities (Fig. 4a). A similar trend was observed in PiperION-based cell (Supplementary Fig. 28), consistent with literature reports[47]. The elevated temperature may accelerate hydrogenation kinetics, leading to a preferential formation of ethylene over other oxygenated products[47]. Regarding stability, the Zirfon 500+ based cell demonstrated stable performance over 250 h of operation at 60 °C (Fig. 4b), whereas the PiperION based cell lasted only around 150 h under identical conditions (Supplementary Figs. 29, 30). In addition, operando EIS of the full cell was performed during the pressure ramping process, and the resulting spectra were analyzed using distribution of relaxation time (DRT) analysis (Supplementary Fig. 31). The Zirfon 500+ based electrolyzer consistently exhibited significantly lower ionic and electronic resistance ($R_1$) throughout the measurement period. Although the slightly higher ohmic resistance of Zirfon 500 +, its superior performance may be attributed to more favorable interfacial contact conditions, which could enhance overall charge transfer efficiency[48]. At 80 °C, both systems exhibited instability within 24 h of operation (Supplementary Fig. 32), mainly attributable to degradation of the cathode (Supplementary Figs. 33–40 and Supplementary Table S2).

When maintaining the temperature at 60 °C while increasing the KOH concentration from 1 M to 6 M, oxygenate selectivity notably improves, particularly for acetate (Fig. 4c). This observation is consistent with what we discovered previously in an AEM-based system, where higher KOH concentrations enhance acetate selectivity due to hydroxide ions facilitating the formation of acetate via reaction with ketene-like intermediates on the copper catalyst[49]. Therefore, transitioning from an AEM to a Zirfon 500+ diaphragm may not alter the mechanism governing acetate formation on the copper catalyst.

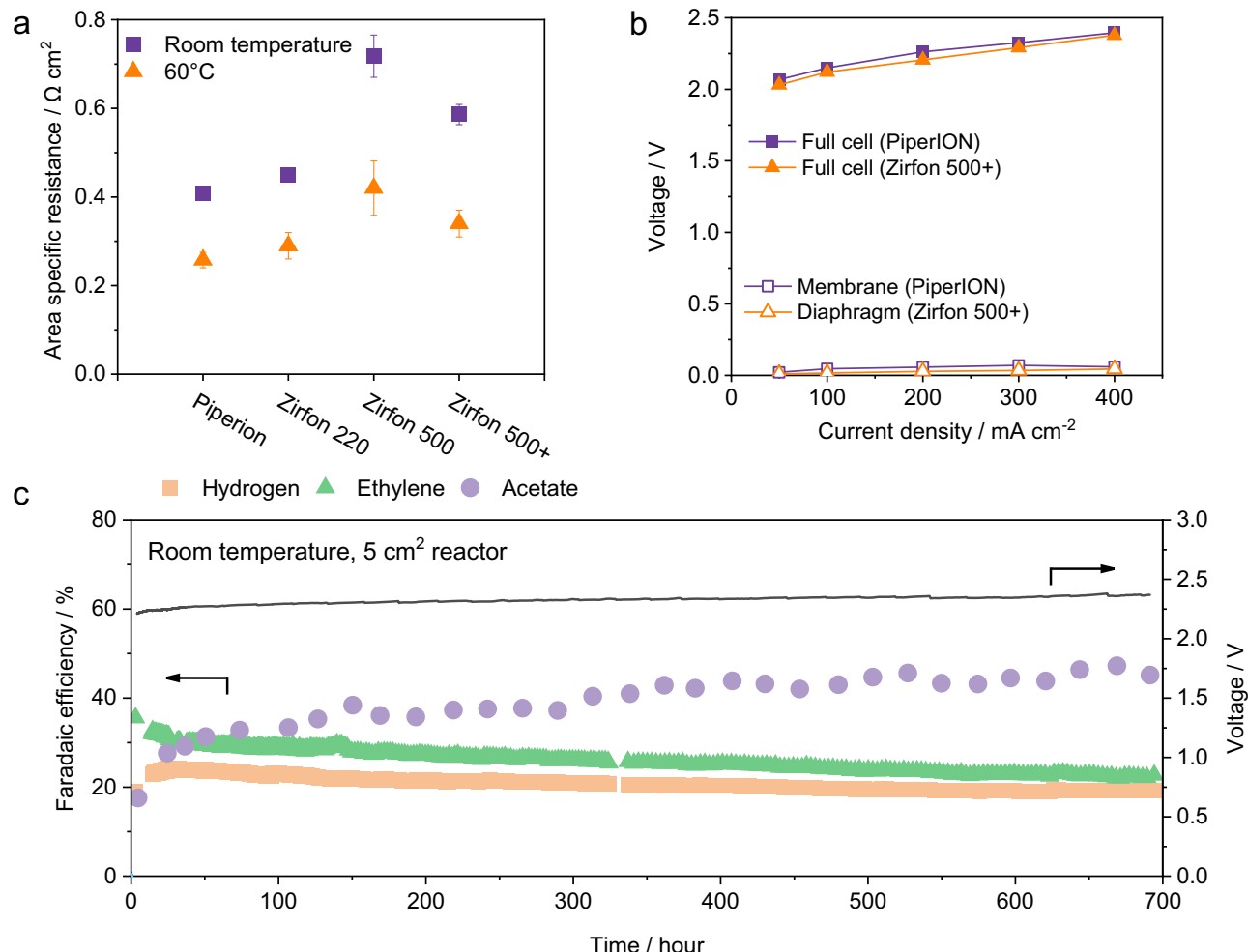

**Fig. 3 | Electrochemical properties of Zirfon materials and their performance in CO electrolysis. a** Measurement of PiperION membrane and Zirfon diaphragms overpotential using a four-electrode flow-through setup (Supplementary Fig. 13). Area-specific resistance of membranes was measured in 1 M KOH using electrochemical impedance spectroscopy (1 mVAC at 0 VDC, 10 steps/decade, $10^5$–10 Hz, 25 °C and 60 °C). **b** Voltammetric performance of full cell (solid symbols) and through-membrane/diaphragm overpotentials (open symbols) for CO electrolysis of PiperION and Zirfon 500+ based system at room temperature in 1 M KOH

anolyte. **c** Stability performance of the COR from Zirfon 500+ based cell: Faradaic efficiencies for acetate, hydrogen, and ethylene, and full cell potential. The cell was operated at a fixed current density of 200 mA cm$^{-2}$ using a 1 M KOH electrolyte, a 40–60 nm Cu nanoparticle cathode, a NiFeO$_x$/Ni foam anode, with CO fed at a rate of 30 sccm at room temperature (∼21 °C). Error bars in (**a**) represent the standard deviation from three independent measurements. All potentials are reported without iR correction. Source data are provided as a Source Data file.

However, we acknowledge that this performance gain comes at the expense of catalyst stability. Our dissolution studies reveal that higher KOH concentrations substantially accelerate Cu degradation (Supplementary Fig. 41), likely due to the enhanced formation of soluble $[Cu(OH)_4]^{2-}$ species under strongly alkaline conditions[50]. Therefore, although acetate selectivity is enhanced at high pH, the accompanying increase in Cu dissolution ultimately undermines long-term system stability. This trade-off underscores the importance of carefully balancing electrolyte composition to optimize both performance and durability.

**Scaling up Zirfon-based CO electrolyzers**

To further evaluate the commercial potential and robustness of the Zirfon-based cell configuration, we scaled up the electrode area from 5 cm$^2$ to 100 cm$^2$ (Supplementary Fig. 1b and 42). Such scaling up effort is a critical step to de-risking the CO electrolysis technology by demonstrating stable and efficient operation under conditions closer to industrial relevance[5,51]. A photograph of the scaled-up CO electrolyzer system is presented in Fig. 5a. The 100 cm$^2$ Zirfon 500+ based cell

was operated with 1 M KOH electrolyte, at room temperature, and at a constant current density of 200 mA cm$^{-2}$. Performance data are illustrated in Fig. 5b, showing a stable operation over 700 h, significantly better than the 100 cm$^2$ PiperION based cell (Supplementary Figs. 43, 44). The full cell potential of the 100 cm$^2$ Zirfon-based cell remained steady at approximately 2.3 V throughout the whole 700 h, which is in good agreement with what we observed at the 5 cm$^2$ cell. In terms of product selectivity, acetate FE is initially 35%, quickly rising and stabilizing at about 50% after extended operation. Ethylene FE, however, exhibited a gradual decline from an initial 32% to approximately 10% after 700 h. Hydrogen FE remained consistently within the range of 25–30% over the entire test duration. In addition, during the assembly of the 100 cm$^2$ electrolyzer, it was found that maintaining the fully hydrated state required for wet assembly of the PiperION membrane posed a considerable engineering challenge. To evaluate the impact of assembly conditions on cell performance, both PiperION and Zirfon 500+ membranes were tested under dry and wet assembly conditions (Supplementary Fig. 45). The results showed that Zirfon 500+ maintained comparable cell selectivity under both assembly approaches,

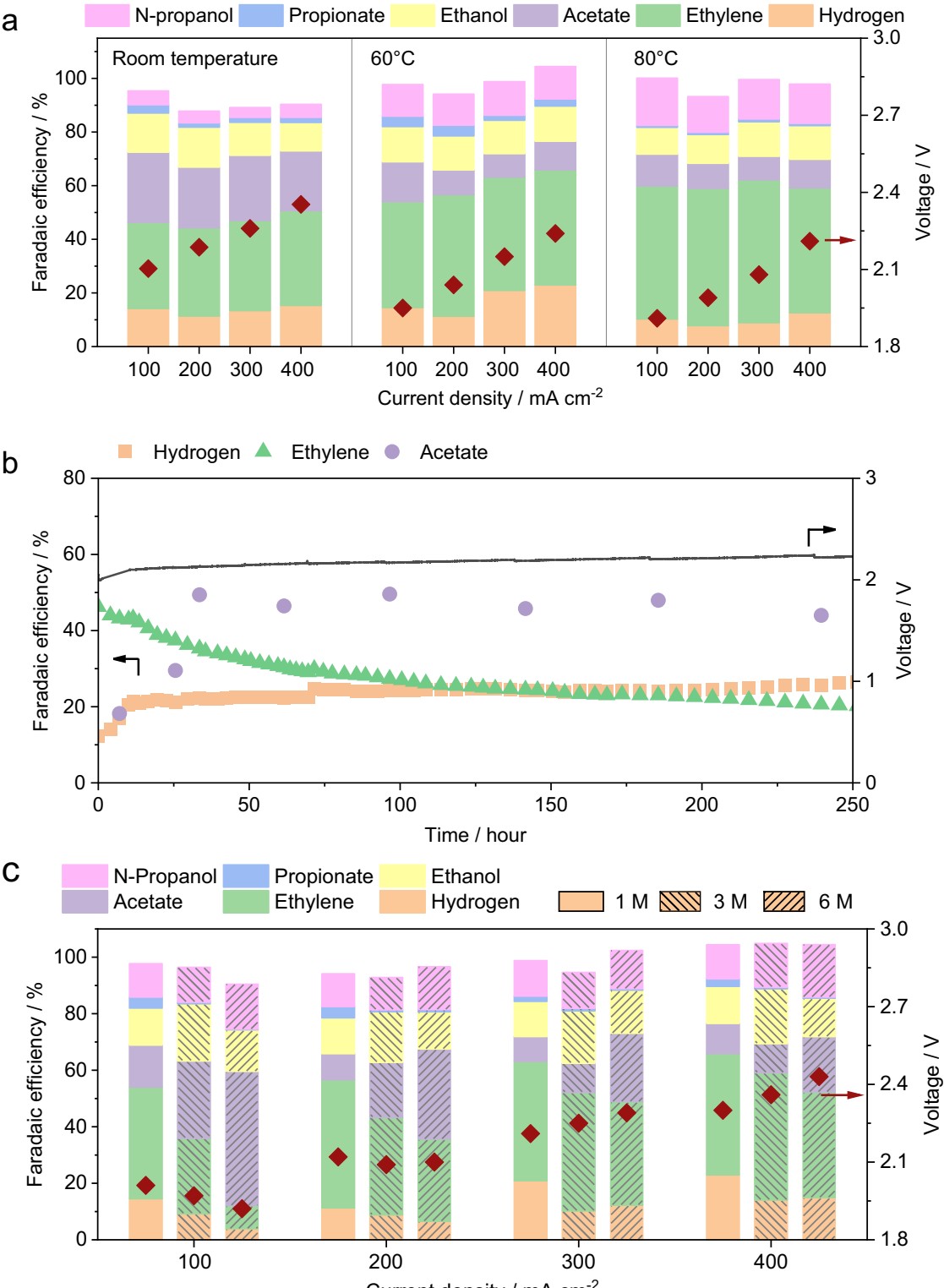

**Fig. 4 | Electrochemical properties of Zirfon 500+ at elevated temperatures.**
**a** Temperature-dependent tests of Zirfon 500+ among different current density.
**b** Stability performance of Zirfon 500+ at 60 °C with a fixed current density of
200 mA cm⁻². **c** Faradaic efficiency for all detectable products and corresponding
cell voltages of Zirfon 500+ based cell measured with different KOH concentrations
(1 M, 3 M, and 6 M). While increasing KOH concentration generally reduces cell

voltage at low current densities by improving ionic conductivity, the opposite trend
is observed at high current densities due to mass transport limitations and inter-
facial effects[62]. The cell was operated at a fixed current density of 200 mA cm⁻² at
60 °C, a 40–60 nm Cu nanoparticle cathode, a NiFeOₓ/Ni foam anode, with CO fed
at a rate of 50 sccm. All potentials are reported without iR correction. Source data
are provided as a Source Data file.

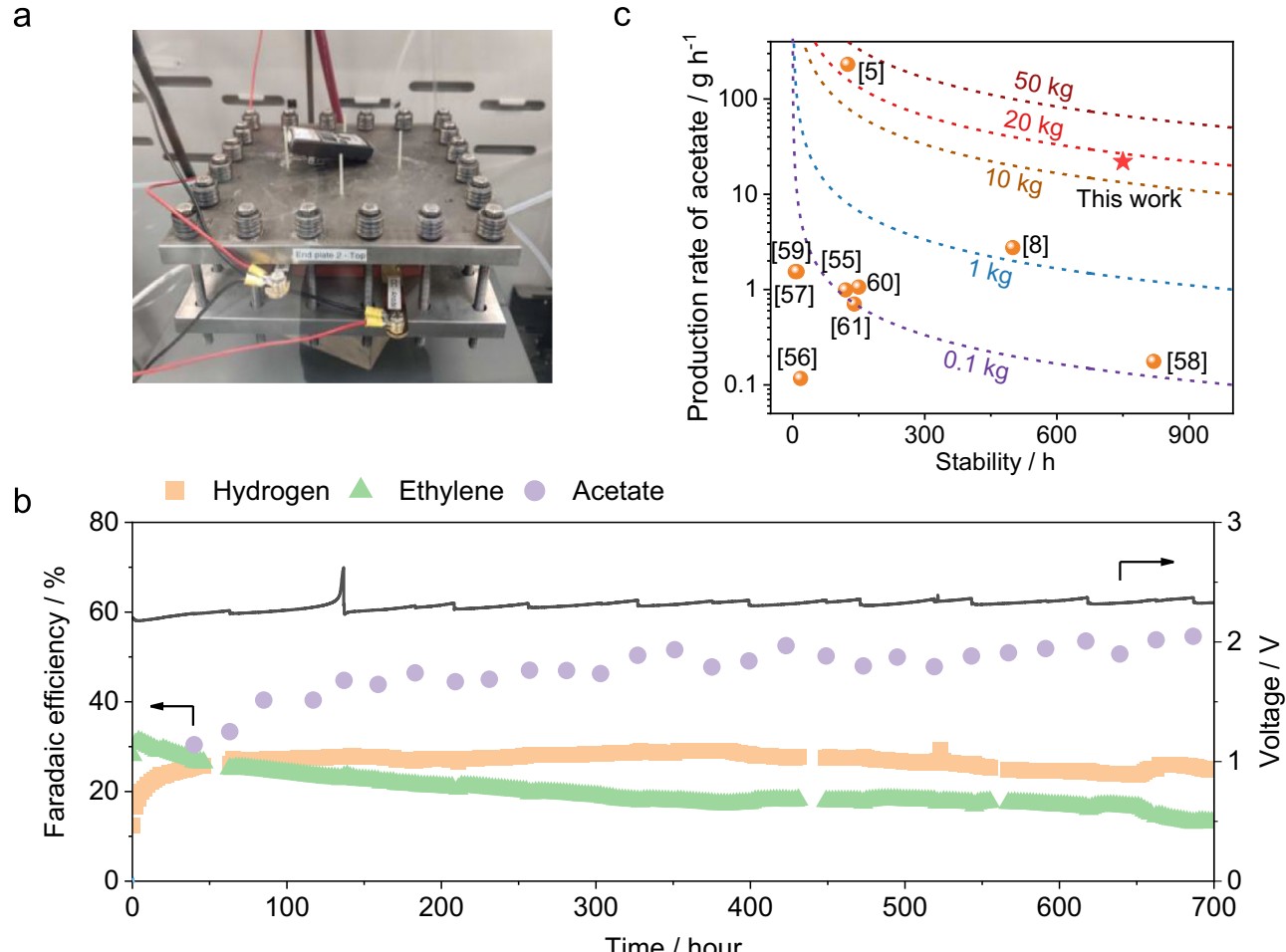

**Fig. 5 | Performance of 100 cm² CO Electrolysis Cell. a** Photograph of 100 cm² CO electrolyzer. **b** Faradaic efficiency of acetate, ethylene and hydrogen and corresponding cell voltages measured using Zirfon 500+ at a fixed current density of 200 mA cm⁻². The 100 cm² zero-gap CO electrolyzer was operated with a 1 M KOH electrolyte at a flow rate of 60 mL min⁻¹, a 40–60 nm Cu nanoparticle cathode, a NiFeOₓ/Ni foam anode, with CO fed at a rate of 400 sccm and at room temperature. The sudden rise in cell voltage was caused by a pH drop in the electrolyte due to acetate buildup during extended electrolysis. Replacing the electrolyte restored the pH and lowered the cell voltage accordingly. **c** Comparison of acetate production rate and operational stability among state-of-the-art CO electrolyzers. Dashed lines represent the total acetate yield achieved during the whole reported operation period[5,8,55–61]. All potentials are reported without iR correction. Source data are provided as a Source Data file.

whereas the PiperION based cell exhibited a significant decline in performance when assembled under non-wet conditions. These findings suggest that the use of Zirfon 500+ can substantially reduce the complexity associated with mechanical assembly in practical electrolyzer stack fabrication.

After 700 h of continuous operation, while the decline in product selectivity could be associated with Cu catalyst restructuring (Supplementary Fig. 46), the Zirfon 500+ based cell experienced gas crossover issues that ultimately led to system failure. Post-reaction SEM and XPS analyses of the Zirfon 500+ and electrode components did not show any obvious structural degradation and exhibited behavior consistent with the degradation patterns observed under ambient and elevated-temperature stability test. (Supplementary Figs. 47–51). In addition, we also observed that a notable change in the surface wettability of the Zirfon 500+ diaphragm was observed, with distinct differences between the cathode-facing and anode-facing sides (Supplementary Fig. 52). Specifically, the cathode-facing side shows a markedly reduced hydrophilicity, exhibiting a contact angle of 80.6°, whereas the anode-facing side retains a similar wettability to the pristine Zirfon 500+ (a low contact angle of 41.9°). The observed decrease in wettability is closely associated with the reduction in $ZrO_2$

content. Once the $ZrO_2$ content drops below 70 wt%—significantly lower than the ~85 wt% in Zirfon—the wettability deteriorates noticeably[18,52–54]. Further analysis revealed a substantial loss of Zr on the cathode side, with the Zr content decreasing by nearly 20 wt% (Supplementary Table S3). Eventually, a pronounced decrease in bubble point was also observed, dropping from an initial 50 psi to 16.2 psi (Supplementary Table S4). This reduction is particularly critical, as the back pressure on the cathode side during testing was set to 17 psi, which correlates well with the experimentally observed gas crossover from cathode to anode during the final stage of electrolysis. this asymmetric deterioration has contributed to the onset of severe gas crossover and subsequent cell failure.

To contextualize our performance, we analyzed the acetate production rates and cell stability reported for CO electrolysis in recent literature[5,8,55–61], as summarized in Fig. 5c. Most existing studies reported a total acetate yield of 0.1 kg or lower. In comparison, our recent 1-kW scale tandem $CO_2$ electrolysis system achieved a total of 28.9 kg acetate production using a combined electrode area of 1000 cm[25]. Remarkably, the current 100 cm² Zirfon-based system has reached a total acetate yield of 16.5 kg over the 700-h continuous test, clearly demonstrating its commercial potential. Future work will focus

on further scaling up the diaphragm-based CO electrolysis system while improving its system robustness under commercial relevant operation conditions. The energy efficiency for carbon products was approximately 2.19%, and the CO utilization was around 9.5% (Supplementary Note 1). The low energy efficiency is primarily attributed to the low thermodynamic potentials of ethylene and acetate relative to the applied cell voltage. The limited CO utilization resulted from the deliberate use of excess CO during long-term operation to maintain carbon product selectivity and reduce experimental complexity. In future studies, we aim to optimize CO flow rates and establish rapid stability screening methods to improve both carbon utilization and testing efficiency.

In summary, this work demonstrates the significant promise of Zirfon 500+ diaphragms for CO electrolysis, particularly in comparison to state-of-the-art AEMs. Zirfon-based electrolyzers exhibited high durability and stable selectivity toward $C_{2+}$ products under various operating conditions, including elevated temperatures and alkaline electrolyte concentrations relevant to commercial applications. Structural characterizations revealed minimal chemical and morphological degradation for Zirfon, unlike AEMs, which showed evident chemical instability. Importantly, successful scale-up from $5\,cm^2$ to $100\,cm^2$ demonstrated stable long-term performance and robust acetate production, significantly outperforming previously reported systems. About $16.5\,kg$ of acetate was produced over $700\,h$ at $200\,mA\,cm^{-2}$ at a $100\,cm^2$ scale, underscoring the commercial viability and scalability of Zirfon-based electrolyzers. Overall, these results highlight the Zirfon diaphragm as a compelling alternative membrane for practical CO electrolysis applications, offering a reliable path forward toward industrial-scale carbon utilization technologies.

## Methods

### Electrode preparation

Cathodes were fabricated by air-spraying cathode ink onto carbon paper (Sigracet 39BB, Fuel Cell Store). A total of 200 mg of commercial Cu (Cu nanopowders, 40–60 nm particle size, ≥99.5%, Sigma-Aldrich) was dispersed with 10 wt.% PTFE Nanopowder (100-150 nm, Nanoshel-UK Ltd.) and 10 wt% Nafion ionomer relative to Cu, the latter introduced via a 20 wt% Nafion ionomer solution (D2021 Nafion Dispersion, Ion Power, Inc.), in 40 mL of a 1:1 mixture of isopropyl alcohol ( ≥ 99.5%, ACS Reagent, Sigma-Aldrich) and Milli-Q water. The catalyst ink was sonicated for at least 30 min below 40 °C to ensure uniform dispersion before being spray-cast onto carbon paper taped onto a hotplate at 100 °C, achieving a final catalyst loading of $1.25\,mg\,cm^{-2}$.

$NiFeO_x$ anodes were fabricated following a previously reported procedure. Porous nickel foam ( > 99.99%, 1.6 mm thickness, MTI Corporation) was sonicated in 5 M HCl (37%, ACS Reagent, Sigma-Aldrich) for 10 min to dissolve surface nickel oxides, followed by sonication in a 1:1 mixture of ethanol ( ≥ 99.5%, ACS Reagent, Sigma-Aldrich) and Milli-Q water for 10 min to remove residual acid and organic impurities. Finally, the foam was sonicated in Milli-Q water for an additional 10 min. For all electrodeposition experiments, the cleaned nickel foam was used as the working electrode, a thermally platinized high-porosity titanium fiber felt (Fuel Cell Store) was employed as the counter electrode, and an Ag/AgCl electrode (Pine Research) served as the reference electrode. Electrodeposition was performed by immersing the nickel foam in an electrolyte bath containing 3 mM nickel (II) nitrate hexahydrate ($Ni(NO_3)_2{\cdot}6H_2O$, 99%, Sigma-Aldrich) and 3 mM iron (III) nitrate nonahydrate ($Fe(NO_3)_3{\cdot}9H_2O$, 99%, Sigma-Aldrich) A potential of −1.0 V vs. Ag/AgCl was applied using a BioLogic SP-150 potentiostat until a total charge of 58 C was passed. Following deposition, the electrode was thoroughly rinsed with Milli-Q water and dried overnight at 60 °C. To achieve the desired thickness, the $NiFeO_x$ anode was first compressed at 5000 lb using a Carver Bench Top Standard Heated Press (Model 25-12H (3856)).

### Electrochemical electrolyzer preparation

CO electroreduction experiments were conducted in a zero-gap electrolyzer with in-house manufactured serpentine flow channels with a reaction area of $5\,cm^2$ and $100\,cm^2$. To evaluate the performance of various separator materials in zero-gap electrolyzers, a series of diaphragms were tested in 1 M potassium hydroxide (KOH, 85%, ACS Reagent, Sigma-Aldrich) at room temperature ( ~ 21 °C) unless otherwise specified, and assembled in a zero-gap configuration with identical electrode materials and gasket sealing protocols. The diaphragms included Zirfon (UTP 220, UTP 500, UTP 500 + , Agfa), Polyethersulfone (PES, 0.1 μm, 0.22 μm, 0.45 μm, and 1.2 μm pore size, Millipore Express®, hydrophilic), Polyvinylidene Fluoride (PVDF, 0.22 μm pore size, Durapore®, hydrophilic), Nylon (0.2 μm pore size, Millipore Express®, hydrophilic) and fiberglass (0.7 μm pore size, Millipore Express®, hydrophilic). In addition, PiperION AEMs (A60-HCO3, Versogen) were employed for comparative analysis. All AEMs were activated in 1 M KOH for 24 h prior to use.

End plates were fabricated from 316 stainless steel. Teflon PTFE gaskets (1 mil, 3 mil, 5 mil, 10 mil McMaster-Carr) and Silicone Rubber Sheet gaskets (10 mil, McMaster-Carr) were used to ensure proper sealing between the cathode and anode compartments. In the $5\,cm^2$ electrolyzer, 10 mil Teflon PTFE gaskets were applied on both the cathode and anode sides. For the $100\,cm^2$ electrolyzer, a combination of 3 mil and 5 mil Teflon PTFE gaskets was used on the cathode side. On the anode side, 1 mil and 3 mil Teflon PTFE gaskets were combined with a 10 mil Silicone Rubber Sheet gasket to achieve effective sealing of the electrode assemblies. Electrodes were positioned on either side of the conductive membrane or diaphragm, and the $5\,cm^2$ cell was compressed to 20 in-lbs using a torque wrench to ensure optimal sealing; for the $100\,cm^2$ electrolyzer, the assembly followed a stepwise torqueing protocol: 60 in-bls in the first round, followed by three subsequent rounds at 120 in-bls, with the final round applied in a clockwise sequence to evenly distribute the pressure and achieve optimal sealing.

### Electrochemical electrolyzer experiments

For the $5\,cm^2$ electrolyzer, CO gas was supplied to the cathode end plate using a mass flow controller (GF040, Brooks Instruments), while gas system pressure was maintained at 16.5 psia using a digital back-pressure controller (Alicat). The 1 M KOH anolyte was delivered by a peristaltic pump (Cole-Parmer) at a constant rate of $3\,mL\,min^{-1}$. Electrochemical measurements were conducted using a BioLogic SP-150 potentiostat equipped with a 10 A booster in a two-electrode configuration or a power supply (Kiprim DC605S variable DC programmable power supply). For different current density experiments, the cell was allowed to run at each applied current for 15 min, after which the anolyte was collected for 2 min. Gas-phase products were analyzed with an inline gas chromatograph (SRI Instruments) equipped with a HayeSep D column, using high-purity argon (99.999%) as the carrier gas. Outlet gas flow rates were measured with an ADM flow meter (Agilent). Liquid products were analysed by $^1$H nuclear magnetic resonance (Agilent DD2, 500 MHz). 400 μl of sample was mixed with 200 μl of 20 mM dimethyl sulfoxide (99.9%, Alfa Aesar) in $D_2O$, which was used as the internal standard. The experimental temperature was regulated by placing the electrolyzer and electrolyte solution in a temperature-controlled oven. The temperature data was collected by thermocouple temperature data logger (OM-HL-EH-TC, Omega Engineering) through insulated thermocouples sticked to the cell.

For continuous stability operation experiments, the CO gas flow rate was set to $30\,mL\,min^{-1}$, while the anolyte (1 M KOH, 200 mL) was continuously recirculated. The anolyte was replaced approximately every 24 h and liquid samples were collected either every 12 h or 24 h. Liquid samples were collected after switching to a fresh anolyte in single-pass mode for 10 min and diluted by 1/20 using deionized water

before being tested by NMR. After sample collection, the anolyte was returned to recirculation mode. The electrochemical cell was operated in galvanostatic mode at 1 A (200 mA cm$^{-2}$). The gas and liquid products were analyzed following the same procedures as those used in the experiments conducted under varying current densities. The 100 cm$^2$ electrolyzer was operated under a constant current density of 200 mA cm$^{-2}$. The CO gas flow rate was set to 400 mL min$^{-1}$, and the electrolyte (1 M KOH) was circulated at a flow rate of 60 mL min$^{-1}$. A Land battery testing system (CT5002A, LANHE) was used to supply current to the electrolyzer. Liquid-phase products were diluted to 0.1 M potassium concentration for analysis via high-performance liquid chromatography (HPLC, Agilent 1260 series). The liquid-phase products were separated using an Aminex HPX-87H column (Bio-Rad) maintained at 50 °C, with detection via a refractive index detector. A 5 mM H$_2$SO$_4$ solution served as the mobile phase at a flow rate of 0.3 mL min$^{-1}$.

All dry and wet assembly experiments were conducted in a 5 cm$^2$ zero-gap CO electrolyzer. In the dry assembly procedure, the activated PiperION or Zirfon 500+ were rapidly air-dried until their dimensions and weight remained constant, and were then assembled into the cell for COR test. The wet assembly procedure refers to the standard process, in which the activated membranes were assembled in their fully hydrated state without prior drying. For PiperION, an additional condition, wet assembly with delayed operation, was also evaluated. In this case, the activated membrane was immediately assembled into the electrolyzer in a wet state, but the system was left idle without introducing electrolyte or CO gas for 5 h prior to initiating COR operation. During all tests, the anolyte consisted of 1 M KOH supplied at 3 mL min$^{-1}$, and the CO gas flow rate was maintained at 30 mL min$^{-1}$.

## Gas crossover experiments

For the quantification of gas crossover-related Faradaic efficiency and volumetric composition, a well-sealed trap was used to contain the anolyte. Argon was employed as a carrier gas to purge the gas products dissolved or released in the anolyte, which were then directed into a gas chromatograph for quantitative analysis.

To further verify the presence of hydrogen gas crossover, a dual-cell system was employed. As illustrated in Supplementary Fig. 10, a hydrogen oxidation reaction (HOR) reactor was assembled downstream of a given COR reactor, with identical cathode, anode, and membrane configurations. The cathodic gas outlet from the COR reactor was directed into the cathode of the HOR reactor, while the anolyte in the HOR reactor was the same as that used in the COR reactor. A potential (1.48 V) slightly lower than the OCV of the COR reactor (~1.52 V) was applied to the HOR reactor. If hydrogen crossover occurred through the tested diaphragm or AEM, the hydrogen reaching the anode side would undergo oxidation due to the lower overpotential of HOR compared to OER, thereby generating a measurable current. The presence of such current served as an indicator of H$_2$ gas crossover under the tested membrane conditions.

The bubble point test was operated with 5 cm$^2$ cell. The diaphragm was inserted between two 1 mil PTFE gaskets and compressed at 35 psi between the endplates. In the experimental setup, deionized (DI) water was introduced on one side of the cell, while argon gas was fed into the other side. The DI water outlet was open to atmospheric pressure, while the backpressure on the gas side was incrementally increased using a digital back-pressure controller to establish a controlled pressure gradient across the membrane. Each pressure step was held for five minutes to ensure system equilibration. The pressure at which gas bubbles were first observed entering the liquid-phase flow channel was recorded as the bubble point.

## Electrochemical impedance spectroscopy experiments

The five-electrode test were conducted as described previously. All experiments were conducted in a 1 M KOH with a 3 mL min$^{-1}$ anolyte

flow using a zero-gap electrolyzer. Each side of the 5 cm$^2$ electrochemical cell consisted of two 5 mil PTFE gaskets: one with an opening to introduce nickel foams to the diaphragm or membrane, and one without an opening and the cell was compressed at 35 psi between the endplates. The electrochemical cell is connected to three reference electrodes using a BioLogic SP-150 potentiostat: two quasi-reference electrodes pressed onto the diaphragm or membrane introduced from cutting openings in a gasket on each side (Supplementary Fig. 13) and a leakless Hg/HgO electrodes reference electrode (Koslow Scientific, 5088 series, standard 1 molar solution) in a custom PTFE holder in the upstream anolyte tubing. Electrodes are deliberately misaligned by cutting corners (2.5 mm × 2.5 mm, 0.031 cm$^2$) to create leading edges to improve reproducibility and minimize transmembrane potential measurement error. Despite the reduced active area (0.6 %), the original area (5 cm$^2$) is used for all geometric normalization calculations. All experiments were operated under galvanostatic conditions at various current densities. After stabilizing at each current density for 10 min, open circuit voltage (OCV) and galvanostatic electrochemical impedance spectroscopy (GEIS) measurements were performed. GEIS was carried out over a frequency range of 10,000 Hz to 1 Hz with 10 points per decade and an amplitude of 15% of the applied DC current. The test was done for full cell (cathode v.s. anode), cathode (cathode v.s. cathode Ni foams), anode (anode v.s. anode Ni foams), membrane (cathode Ni foams v.s. anode Ni foams), and anolyte (anode Ni foams v.s. anolyte fritted reference electrode). For alleviated temperature continuous experiments, full cell impedance results were interpreted via distributed relaxation times (DRT) methodology. A Matlab code implementation for DRT fitting was employed for analysis (retrieved from https://sites.google.com/site/drttools/ 2022/01/19).

## Membrane and diaphragm conductivity test

Membrane and diaphragm conductivity test was done using a four-electrode setup. The working and counter electrodes were connected to stainless-steel endplates. Two Hg/HgO electrodes were used as the sensing and reference electrodes. 1 M KOH was bubbled with argon and flown using a peristaltic pump. The diaphragm was inserted between two 1 mil PTFE gaskets and compressed at 35 psi between the endplates. Electrochemical impedance spectroscopy was performed using the frequency response analyzer measurement function in Autolab Nova (1 mVAC at 0 VDC, 10 steps/decade, 10$^5$–10 Hz). An equivalent circuit was fitted to the impedance spectra to determine the high-frequency resistance. The high-frequency resistance was normalized to the active area to obtain the area-specific resistance (ASR). The background ASR was subtracted from the total cell ASR to obtain the diaphragm ASR for each diaphragm. Diaphragm conductivity was obtained by normalizing the diaphragm ASR by the dry thickness.

## Material characterization

Sample morphology was characterized using a scanning electron microscope (Thermofisher Quattro S Environmental Scanning Electron Microscope) before and after test. To minimize oxygen exposure and preserve the Cu structure, all post-samples were prepared and stored in an Ar-filled glovebox immediately after cell disassembly and cathode washing with Milli-Q water. The samples were then dried in the glovebox transfer chamber under vacuum conditions for 24 h. To further limit air exposure, the samples were transferred from the glovebox to the SEM equipment using an argon-purged plastic bag, ensuring total exposure remained under 30 seconds throughout the process. Imaging was conducted at an accelerating voltage of 5.00 kV with a working distance of 8 mm, utilizing secondary electron and concentric backscatter detectors to capture high-resolution surface images. Images for energy dispersive X-ray analysis were taken at a voltage of 10 kV. XPS was completed on samples before and after test. All pre-samples were exposed to air due to the storage of catalysts in ambient conditions. All post-samples were preserved using the same

protocol as that used for SEM analysis. XPS equipment (Physical Electronics 5000 VersaProbe II Scanning ESCA Microprobe) was used to obtain high-resolution XPS measurements at a pass energy of 20 eV with a step size of 0.1 eV. FTIR spectra were obtained using a Thermo Scientific Nicolet iS20 FTIR spectrometer with an attenuated total reflection (ATR) accessory. The spectra were recorded with 0.125 cm$^{-1}$ intervals at room temperature. Each spectrum was averaged over 32 scans. All samples were rinsed thoroughly with Milli-Q water, vacuum-dried, and analyzed directly without further treatment.

Inductively coupled plasma mass spectrometry was completed using a NexION 2000 PerkinElmer. Samples were neutralized with nitric acid (70%, >99.999% trace metals basis, Sigma-Aldrich) then diluted 200 times using 1 wt% nitric acid solution. Standards were produced via serial dilution of 10 ppm trace metal standard (IV-ICPMS-71A, Inorganic Ventures) in 1 wt% nitric acid. Standards ranged from 1 ppm to 50 ppm.

## Data availability
All the data that support the findings of this study are available within the paper and its Supplementary Information files, or from the corresponding author on reasonable request. Source data are provided with this paper.

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

## Acknowledgements
We acknowledge financial support from the Gates Foundation (INV-051757) and the National Science Foundation (Award number: EEC-2330245). The authors also thank Wei Wei for assistance with the contact angle measurements, Ahryeon Lee for assistance with SEM analysis, Dr. Weiwei Li for sample preparation, and Dr. Gregery S. Hutchings for thoughtful discussions. The authors acknowledge financial support from Washington University in St. Louis and the Institute of Materials Science & Engineering for the use of instruments and staff assistance.

## Author contributions
W.D. and S.X. conducted most of the experiments and contributed equally to this work. G.W.P.M. assisted with the five-electrode measurements. Z.W. contributed to sample preparation. B.S.C. collected the liquid samples during the 5 cm² cell stability tests. F.J. conceived the idea and supervised the whole project. All authors reviewed and commented on the final version of the paper.

## Competing interests
B.S.C. is associate director of technical strategy and F.J. is co-founder and scientific advisor at Lectrolyst, a company that develops $CO_2$ electrolysis technologies. The remaining authors declare no competing interests.
