## [Transparent Peer Review file · Nature Communications]

Diaphragm-based Carbon Monoxide Electrolyzers for Multicarbon Production under Alkaline Conditions

Corresponding Author: Professor Feng Jiao

Version 0:

Reviewer comments:

Reviewer #1

(Remarks to the Author)

The manuscript presents a timely and technically well-executed study on the use of Zirfon diaphragms as low-cost, chemically robust alternatives to AEMs for CO electrolysis. The authors systematically evaluate a range of diaphragm materials and demonstrate that Zirfon 500+ not only delivers comparable FEs for acetate but also exhibits superior long-term stability under elevated temperatures and during scale-up to 100 cm² cell area. The work is of high relevance to the electrosynthesis and CO₂ utilization communities, particularly in light of the ongoing push for scalable and durable electrochemical conversion systems. However, several technical issues warrant further discussion or clarification. First, the mechanisms underlying Zirfon degradation, particularly the observed ZrO₂ loss and asymmetric wettability shifts, remain unclear and should be addressed with deeper materials analysis. Second, while scale-up is demonstrated, the manuscript lacks discussion of energy efficiency, CO utilization, and other system-level metrics necessary to evaluate commercial viability. Lastly, the impact of diaphragm aging on gas permeability, ionic transport, and mechanical integrity over time should be more rigorously assessed. Overall, this study makes a valuable contribution by expanding the material options for CO electrolysis and providing evidence for diaphragm-based system feasibility. I recommend its publication in Nature Communications after addressing below concerns.

Specific issues:

1. The gas crossover behavior changes over longer-term aging is not discussed. In addition, while bubble point test is informative, it does not account for dynamic effects such as pressure fluctuations, wetting behavior, or mechanical deformation under operating conditions.
2. The authors report that Zirfon 500+ has a higher area-specific resistance than PiperION yet exhibits a lower full-cell voltage across all current densities. While a voltage breakdown is provided, the explanation remains qualitative. The manuscript should quantify the relative contributions of cathodic kinetics (e.g., Tafel slopes) or interfacial resistances to support this observation.
3. The disappearance of the N 1s peak in PiperION is used to infer membrane degradation. However, XPS is surface-sensitive and prone to sampling artifacts, especially when exposed to electrolyte residues. The manuscript should clarify whether thorough cleaning was performed, and ideally support XPS results with additional characterization (e.g., FTIR) to validate chemical degradation.
4. The authors report a decline in C₂+ selectivity over 700 hours in the 100 cm² cell. This shift is not correlated with any catalyst metrics. Without a time-resolved analysis of the local environment or cathode state (e.g., Cu speciation), it is difficult to decouple catalyst deactivation from diaphragm-driven effects. Similarly, the failure of both cells at 80 °C is attributed to cathode degradation, yet supporting evidence is limited. What specific catalyst degradation pathways occur? Do these pathways differ between AEM and diaphragm systems due to membrane-electrode interactions?
5. While increased KOH concentration is linked to higher acetate selectivity via ketene stabilization, its possible acceleration of Zirfon degradation is not discussed. Given that ZrO₂ solubility can increase in highly alkaline environments, especially under cathodic polarization, further control experiments at varied pH would help clarify trade-offs between performance and durability.

Reviewer #2

(Remarks to the Author)

This manuscript reports a diaphragm-based carbon monoxide (CO) electrolyzer for multicarbon production, which replaces current unstable, high-cost AEMs with low-cost, durable Zirfon-based diaphragm. Authors were able to successfully operate 5-cm² electrolyzers at 200 mA/cm² for over 700 hours, whereas current AEM-based electrolyzers were only able to operate around ~150 hours, demonstrating highly enhanced stability. The system was further scaled up to 100-cm² cells and were also operated at 200 mA/cm² for over 700 hours, demonstrating a total acetate yield of 16.5 kg, demonstrating its commercial potential. In overall, the manuscript is well-written, and replacing the AEMs which has been considered as the key limiting factor for most alkaline-environment operating electrochemical cells with robust, cheap materials is of importance. Furthermore, the demonstration of a practical large-scale device is considered as a major breakthrough in the field. Therefore, the reviewer thinks this manuscript should be published with minor revisions.

Comments 1.: Porous diaphragms can allow the direct anolyte crossover to the cathode side, since they possess large, abundant pores. This can result in a few problems:

(a) The purity of CO-reduction product can be effected. Authors should conduct K⁺ IC and pH analysis of the cathode outlet product for both diaphragm and membrane-based systems and compare their product purity.

(b) Liquid crossover can lead to the flooding of the cathode chamber, even though gas is constantly supplied. Given that the diaphragms are relatively undamaged after long-term stability tests; could this be the reason to the final degradation of cells after 700+ hours?

Comments 2.: Does utilizing diaphragms with higher porosities enable more CO to crossover to the anode side and negatively impact the single-pass conversion efficiency of the system? It would be better if the authors could detect the content of CO at the anolyte outlet and calculate the SPCEs for both diaphragm and membrane-based systems.

Comments 3.: In Fig.4(c), the cell potentials for 3M and 6M conditions are lower than 1M condition at lower current densities, but is reversed at high current densities. Considering that higher ionic conductivity can be provided with concentrated electrolytes, what could be the reason for this?

Comments 4.: Is there a specific reason that the authors applied this diaphragm-based systems to CO electrolysis? Would it be possible to demonstrate a CO₂ reduction electrolyzer with this system?

Comments 5.: In overall, there are some formatting errors in Page 12, 17 in manuscript and Page 28, 34 of supporting document.

Reviewer #3

(Remarks to the Author)

This manuscript systematically investigated a series of diaphragms as alternative separators for alkaline CO electrolysis and compared them with alkaline polyelectrolyte membranes (AEMs). The CO electrolysis performance using some of these diaphragms was comparable to that of state-of-the-art AEMs at current densities ranging from 50 to 400 mA cm⁻². It is noted that many reports on alkaline CO electrolysis have achieved high current densities up to several A cm⁻². So, a question arises: why did this paper conduct performance evaluations at relatively lower current densities? Admittedly, the performance of the 100 cm² Zirfon-based cell was truly striking. While the paper presented numerous performance comparisons, it lacked an analysis of the reasons for performance changes. For example, could some in-situ spectroscopy methods be used to study the reaction mechanism at the cathode interface, to clarify the influence mechanism of the diaphragm on CORR? The specific comments are as follows:

1. Using XPS survey spectra to analyze the degradation of PiperION is not a reasonable approach. What is the degradation pathway of piperidine N? It is recommended to conduct a specific analysis on N1s spectra.

2. Regarding the stability tests of Fig. 4 (Zirfon) and the ones using the PiperION membrane in the SI, what are the reasons for the rapid increase in hydrogen evolution in the later stage and the rapid decrease in ethylene in the early stage? I do not think that the rapid performance decline at the initial 20 hours is due to the defects of membrane materials.

3. Why does the cell voltage increase at high current densities as the alkali concentration increases?

4. What causes the sudden increase in cell voltage at 100 h in Fig. 5? In several stability testing plots, the selectivity for acetate was low at the beginning, and the selectivity for alcohol products was not given. Is there a possibility that alcohol products are oxidized to acetate at the anode side?

Version 1:

Reviewer comments:

Reviewer #1

(Remarks to the Author)

Reviewer #2

(Remarks to the Author)

The authors have carefully addressed my comments, and I think this work is ready for publication.

Reviewer #3

(Remarks to the Author)

The authors have addressed all my queries. In my opinion, this manuscript meets the requirements and is worthy of publication.

Point by Point Response to Review Comments

Reviewer: 1

General Comments R1: *The manuscript presents a timely and technically well-executed study on the use of Zirfon diaphragms as low-cost, chemically robust alternatives to AEMs for CO electrolysis. The authors systematically evaluate a range of diaphragm materials and demonstrate that Zirfon 500+ not only delivers comparable FEs for acetate but also exhibits superior long-term stability under elevated temperatures and during scale-up to 100 cm² cell area. The work is of high relevance to the electrosynthesis and CO₂ utilization communities, particularly in light of the ongoing push for scalable and durable electrochemical conversion systems. However, several technical issues warrant further discussion or clarification. First, the mechanisms underlying Zirfon degradation, particularly the observed ZrO₂ loss and asymmetric wettability shifts, remain unclear and should be addressed with deeper materials analysis. Second, while scale-up is demonstrated, the manuscript lacks discussion of energy efficiency, CO utilization, and other system-level metrics necessary to evaluate commercial viability. Lastly, the impact of diaphragm aging on gas permeability, ionic transport, and mechanical integrity over time should be more rigorously assessed. Overall, this study makes a valuable contribution by expanding the material options for CO electrolysis and providing evidence for diaphragm-based system feasibility. I recommend its publication in Nature Communications after addressing below concerns.*

Response: We thank the reviewer for the positive feedback and thoughtful suggestions. We have addressed all concerns as follows:

1. **Zirfon degradation and wettability shifts:**

We agree with the reviewer that the degradation mechanism of Zirfon, particularly ZrO₂ loss and wettability changes, requires further clarification. It is well-established that the wettability of the Zirfon diaphragm is closely tied to its ZrO₂ content. Commercial Zirfon typically contains ~85 wt% ZrO₂, and literature reports (e.g., doi: 10.1016/S0360-3199(97)00069-4) have shown that maintaining ZrO₂ content above 80 wt% is critical for ensuring high hydrophilicity. A decline below 70 wt% significantly compromises water uptake and wettability (doi: 10.1016/j.ijhydene.2020.08.175; 10.1002/pat.3411; 10.1016/j.ijhydene.2011.03.056).

To investigate Zr loss during operation, we conducted a 24-hour electrolysis test in 1 M KOH at 200 mA cm⁻². ICP-MS analysis of electrolyte samples showed detectable Zr dissolution from the early stages, with a total leaching rate of ~0.01 μg h⁻¹ cm⁻². Of this, ~0.0075 μg h⁻¹ cm⁻² accumulated in the anolyte, and ~0.0026 μg h⁻¹ cm⁻² was captured in a downstream cathodic trap. XPS characterization further revealed significant Zr depletion on the cathode-facing side of the diaphragm (65.3 wt%) relative to the anode-facing side (80.2 wt%), supporting the hypothesis that dissolved Zr species primarily migrate across the diaphragm and accumulate in the anolyte. These findings directly correlate ZrO₂ loss with the observed decline in hydrophilicity and provide mechanistic insights into Zirfon degradation. The detailed results and discussion have been added to the revised manuscript (page 12–13 and Supplementary Information, page 57).

2. **System-level metrics:**

In response to the reviewer's suggestion, we have now included key system-level metrics in the revised manuscript. Specifically, the energy efficiency for carbon-based products during the 700-hour test was approximately 2.19%, and the CO utilization efficiency was around 9.5% in the 100 cm² electrolyzer. These values, while modest, are typical for acetate-selective systems under alkaline conditions and highlight the opportunity for future optimization. The discussion has been added to page 13 of the main text and Supplementary Information pages 59–60.

3. **Diaphragm aging effects:**

To assess how long-term operation impacts diaphragm performance, we compared the properties of fresh and aged Zirfon 500+ samples after 700 hours of continuous electrolysis. The aged diaphragm exhibited a ~10% reduction in tensile strength, indicating minor mechanical degradation while retaining functional integrity. However, the hydrophilicity of the cathode-facing surface declined markedly, with the water contact angle increasing from 41.9° to 80.6°, consistent with reduced ZrO₂ content. Electrical resistance also increased, suggesting a decrease in ionic conductivity. Furthermore, the bubble point dropped from 50 psi to 16.2 psi after aging, which is particularly important since the cathode was operated at 17 psi back pressure—close to the new crossover threshold. This change aligns well

with the experimentally observed increase in gas crossover during late-stage testing. These results are now discussed in the revised manuscript (page 12–13 and Supplementary Information, page 58).

We thank the reviewer again for these insightful comments, which have significantly improved the clarity and depth of our manuscript.

Page 12-13:

The observed decrease in wettability is closely associated with the reduction in ZrO₂ content. Once the ZrO₂ content drops below 70 wt%—significantly lower than the ~85 wt% in Zirfon—the wettability deteriorates noticeably.^{18,60-62} Further analysis revealed a substantial loss of Zr on the cathode side, with the Zr content decreasing by nearly 20 wt% (Supplementary Table S3). Eventually, a pronounced decrease in bubble point was also observed, dropping from an initial 50 psi to 16.2 psi (Supplementary Table S4). This reduction is particularly critical, as the back pressure on the cathode side during testing was set to 17 psi, which correlates well with the experimentally observed gas crossover from cathode to anode during the final stage of electrolysis. This asymmetric deterioration has contributed to the onset of severe gas crossover and subsequent cell failure.

Page 13:

The energy efficiency for carbon products was approximately 2.19%, and the CO utilization was around 9.5% (Supplementary Note 1). The low energy efficiency is primarily attributed to the low thermodynamic potentials of ethylene and acetate relative to the applied cell voltage. The limited CO utilization resulted from the deliberate use of excess CO during long-term operation to maintain carbon product selectivity and reduce experimental complexity. In future studies, we aim to optimize CO flow rates and establish rapid stability screening methods to improve both carbon utilization and testing efficiency.

Supplementary Page 57-60:

Supplementary Table S3 | Zr percentage of post-reaction Zirfon 500+ at cathode and anode side after 100 cm² electrolyzer stability test experiments

Zirfon sample	wt% from EDX	Atom% from EDX	wt% from XPS	Atom% from XPS
Pristine	85.0	42.7	83.5	40.32
Post-reaction sample face cathode side	65.3	19.8	50.3	11.78
Post-reaction sample face anode side	80.2	34.8	70.5	23.97

To further elucidate the mechanism underlying Zr loss during in situ CO electrolysis testing, we conducted a 24-hour electrolysis experiment in 1 M KOH electrolyte at a current density of 200 mA cm⁻². Electrolyte samples from the anode side and the downstream trap solution on the cathode side were collected to trace the migration pathway of lost Zr. ICP-MS analysis revealed detectable Zr dissolution even in the early stages of electrolysis, with a total Zr loss rate of approximately 0.01 μg h⁻¹ cm⁻². Specifically, about 0.0075 μg h⁻¹ cm⁻² of Zr was found to migrate into the anolyte, while approximately 0.0026 μg h⁻¹ cm⁻² was captured in the cathodic trap. Based on the post-test XPS analysis, which shows that the Zr content on the cathode-facing side of the Zirfon 500+ (65.3 wt%) is significantly lower than that on the anode-facing side (80.2 wt%), it can be inferred that most of the dissolved Zr on the cathode side diffuses across the diaphragm and accumulates in the anolyte, while only a small portion is carried out by the gas stream and collected in the cathodic trap. These results provide direct evidence that ZrO₂ leaches progressively from the Zirfon 500+ during electrochemical operation, ultimately leading to the degradation of its hydrophilic properties over time.

Supplementary Table S4 | Comparison of Zirfon 500+ properties before and after long-term CO electrolysis

	Cathode-side wettability (degree)	Tensile strength (MPa)	Resistance (Ω cm ²)	Bubble point (psi)
Pristine sample	41.9	25.7	0.59	50

Post reaction sample	80.6	23.8	0.67	16.2
------	------	------	------

To further evaluate the impact of Zirfon 500+ diaphragm aging on its overall performance, we compared key properties of pristine and aged samples after 700 hours of stability testing in a 100 cm² CO electrolysis cell (Supplementary Table S4). The post-test diaphragm exhibited a slight decrease (~10%) in tensile strength compared to the fresh sample, indicating minor mechanical degradation while still retaining adequate structural integrity. Notably, the hydrophilicity of the cathode-facing side declined significantly, with the water contact angle increasing from 41.9° to 80.6°, which aligns with the observed reduction in ZrO₂ content on the cathode side. Additionally, the measured resistance of the diaphragm increased after long-term testing, suggesting that electrolyte transport through the diaphragm became more restricted. A pronounced decrease in bubble point was also observed, dropping from an initial 50 psi to 16.2 psi. This reduction is particularly critical, as the back pressure on the cathode side during testing was set to 17 psi, which correlates well with the experimentally observed gas crossover from cathode to anode during the final stage of electrolysis.

Supplementary Note 1 | Energy efficiency and CO utilization

1. Energy Efficiency (EE)

The energy efficiency (EE) for each product was calculated using the following equation:

$$EE = (E^\circ / E_{\text{cell}}) \times FE$$

where E° is the standard thermodynamic potential of the product, E_{cell} is the full cell voltage (2.2 V in our case), and FE is the Faradaic efficiency of the product.

For acetate and ethylene, the standard potential is approximately 0.08 V vs. RHE. Based on the long-term stability data, the average Faradaic efficiencies were ~45% for acetate and ~15% for ethylene.

Thus:

$$\text{- EE for acetate: } (0.08 / 2.2) \times 45\% = 1.64\%$$

$$\text{- EE for ethylene: } (0.08 / 2.2) \times 15\% = 0.55\%$$

- Total energy efficiency from carbon products: $EE_{\text{total}} = 1.64\% + 0.55\% = 2.19\%$

2. CO Utilization Efficiency

The CO inlet flow rate was 400 sccm, equivalent to 0.4 L/min. Under standard conditions (22.4 L/mol), this corresponds to:

$$CO_{\text{in}} = 0.4 / 22.4 \approx 0.01786 \text{ mol/min}$$

To estimate how much CO was consumed, we first calculated the total number of electrons transferred based on the assumed operating current density of 200 mA/cm² over a 100 cm² electrode area:

$$I = 200 \times 100 = 20 \text{ A}$$

$$n_e = 20 / 96485 \approx 0.01244 \text{ mol e}^-/\text{min}$$

Using the number of electrons required per product molecule (8 e⁻ for acetate, 12 e⁻ for ethylene), and 45% FE for acetate and 15% FE for ethylene, the product formation rates are:

$$\text{- Acetate: } (0.01244 \times 0.45) / 8 \approx 0.00070 \text{ mol/min}$$

$$\text{- Ethylene: } (0.01244 \times 0.15) / 12 \approx 0.00016 \text{ mol/min}$$

Corresponding CO consumption (2 mol CO per molecule of acetate or ethylene):

$$\text{- Acetate: } 2 \times 0.00070 = 0.0014 \text{ mol CO/min}$$

$$\text{- Ethylene: } 2 \times 0.00016 = 0.00032 \text{ mol CO/min}$$

$$\text{- Total CO consumed} \approx 0.0017 \text{ mol/min}$$

Therefore:

$$CO \text{ Utilization} = (0.0017 / 0.01786) \times 100\% \approx 9.5\%$$

Specific Comment R1-1: *The gas crossover behavior changes over longer-term aging is not discussed. In addition, while bubble point test is informative, it does not account for dynamic effects such as pressure fluctuations, wetting behavior, or mechanical deformation under operating conditions.*

Response: We appreciate the reviewer's suggestion. Same reply in the R1 of diaphragm aging effects.

Specific Comment R1-2: *The authors report that Zirfon 500+ has a higher area-specific resistance than PiperION yet exhibits a lower full-cell voltage across all current densities. While a voltage breakdown is provided, the explanation remains qualitative. The manuscript should quantify the relative contributions of cathodic kinetics (e.g., Tafel slopes) or interfacial resistances to support this observation.*

Response: We thank the reviewer for raising this important point. Voltage decomposition revealed that the diaphragm ohmic resistance in the Zirfon-based cell was indeed higher than that in the PiperION-based cell. However, its contribution to the total cell voltage was minimal. This is because, in 1 M KOH electrolyte, the non-ohmic and thermodynamic potentials at the cathode and anode dominate the total cell voltage, accounting for over 95% of the total contribution, rendering the transmembrane ohmic resistance relatively negligible (Supplementary Fig. 16). Therefore, the difference in membrane resistance itself does not lead to a substantial impact on the overall energy efficiency of the system.

Interestingly, during the analysis, we observed that the cathodic non-ohmic potential in the Zirfon-based cell was consistently lower than that in the PiperION-based cell across all current densities (Supplementary Fig. 17c,d). This could be attributed to the higher compressibility of Zirfon, which may allow for better interfacial contact with the cathode, thereby reducing contact resistance and lowering the cathodic non-ohmic potential. In contrast, the anodic non-ohmic potential was consistently higher in the Zirfon-based cell (Supplementary Fig. 17e,f). On the anode side, the larger water contact angle of Zirfon compared to PiperION (Supplementary Fig. 19) may facilitate the accumulation of gas bubbles at the diaphragm–anode interface, leading to a higher anodic non-ohmic potential.

In summary, a higher area-specific resistance of Zirfon does not significantly impact the overall cell voltage. Instead, the interfacial contact differences introduced by its physical properties in the MEA system appear to play a more prominent role. Therefore, future efforts to modify the diaphragm–electrode interface may offer a promising direction for further optimization, although this lies beyond the scope of the present study.

Regarding the cathodic kinetics (e.g., Tafel slopes) suggested by the reviewer, we acknowledge that due to technical limitations of the zero-gap cell configuration, it is challenging to accurately measure Tafel slopes. This is primarily because inserting a reference electrode to determine the cathode potential is not straightforward in this setup.

We have included these explanations in the revised manuscript (page 8 and supplementary page 18 and 21).

Page 8:

The analysis revealed that the membrane ohmic resistance accounts for only a small fraction of the total cell voltage, with over 95% of the voltage drop arising from the cathodic and anodic non-ohmic and thermodynamic components (Supplementary Fig. 16-19).

Supplementary Page 18:

The Zirfon-based cell exhibited a lower cathodic non-ohmic potential (Supplementary Fig. 17), which can be attributed to the higher compressibility of the Zirfon diaphragm (Supplementary Fig. 18), allowing better interfacial contact with the cathode and thereby reducing contact resistance.² In contrast, the anodic non-ohmic potential was slightly higher in the Zirfon-based cell (Supplementary Fig. 17), possibly due to its larger water contact angle (Supplementary Fig. 19), which may promote gas bubble accumulation at the diaphragm–anode interface and increase interfacial resistance. In summary, a higher area-specific resistance of Zirfon does not significantly impact the overall cell voltage. Instead, the interfacial contact differences introduced by its physical properties in the MEA system appear to play a more prominent role.

Supplementary Page 21:

Supplementary Fig. 19 | Water contact angle measurements on (a) fresh Zirfon 500+ and (b) activated PiperION membranes before and immediately upon water droplet contact.

When a water droplet was placed on fresh Zirfon 500+, it exhibited a contact angle of approximately 41°, whereas on the activated PiperION membrane, the droplet was instantly and completely absorbed, preventing any measurable contact angle. This indicates that the activated PiperION possesses significantly superior hydrophilicity compared to Zirfon 500+.

Specific Comment R1-3: *The disappearance of the N 1s peak in PiperION is used to infer membrane degradation. However, XPS is surface-sensitive and prone to sampling artifacts, especially when exposed to electrolyte residues. The manuscript should clarify whether thorough cleaning was performed, and ideally support XPS results with additional characterization (e.g., FTIR) to validate chemical degradation.*

Response: We thank the reviewer for highlighting the limitations of XPS as a surface-sensitive technique and for suggesting further validation of membrane degradation.

To minimize sampling artifacts, we took great care in post-electrolysis sample preparation. Immediately after electrolysis, the PiperION membrane was removed from the cell and thoroughly rinsed with copious amounts of Milli-Q water to eliminate any residual electrolyte and surface-adsorbed species. The membrane was then dried under vacuum in a desiccator until no visible moisture remained before surface characterization.

To assess the chemical degradation of PiperION, we employed both XPS and FTIR. XPS analysis revealed a substantial decrease in surface nitrogen content, with the N 1s signal becoming undetectable after 120 hours of CO electrolysis at 200 mA cm⁻². While we acknowledge that XPS alone cannot rule out surface contamination effects, the complete disappearance of nitrogen is strongly indicative of degradation or loss of quaternary ammonium functional groups.

To further corroborate these findings, we performed FTIR spectroscopy on the same membrane samples. The FTIR spectra revealed three key changes:

1. **Reduced intensity in the 1450–1500 cm⁻¹ region**, corresponding to C–N bending and N–CH₃ wagging vibrations associated with piperidinium-based quaternary ammonium groups (doi: 10.1177/0892705712471359). This suggests a breakdown of the ion-conducting groups.
2. **Disappearance of the ~1330 cm⁻¹ peak**, assigned to the symmetric bending of methyl groups in N⁺(CH₃)₂ moieties (doi: 10.1016/j.molliq.2022.120484), further supporting the decomposition of quaternary ammonium functionalities.
3. **Emergence of a new peak at ~1700 cm⁻¹**, characteristic of C=O stretching vibrations. This is commonly associated with carbonyl-containing degradation products, including aldehydes and ketones, and suggests oxidative degradation of the polymer backbone (doi: 10.1016/S0141-3910(96)00218-2).

These observations point to a degradation mechanism involving both the loss of functional groups and oxidative breakdown of the polymer matrix. The results collectively support the conclusion that PiperION membranes undergo chemical degradation under prolonged CO electrolysis conditions, beyond surface-level effects.

We have included these explanations in the revised manuscript (page 8, 22 and Supplementary page 28-29).

Page 8:

Additionally, XPS and FTIR revealed the disappearance of the N 1s peak initially present in the pristine PiperION membrane (Supplementary Fig. 26 and 27), suggesting the degradation of PiperION during COR.

Page 22:

FTIR spectra were obtained using a Thermo Scientific Nicolet iS20 FTIR spectrometer with an attenuated total reflection (ATR) accessory. The spectra were recorded with 0.125 cm⁻¹ intervals at room temperature. Each spectrum was averaged over 32 scans. All samples were rinsed thoroughly with Milli-Q water, vacuum-dried, and analyzed directly without further treatment.

Supplementary Page 28-29:

XPS experimental sample pretreat: After completion of the electrolysis, the AEM was carefully removed from the electrochemical cell and immediately rinsed with a large volume of Milli-Q water to remove any residual electrolyte from the membrane surface. Following the rinsing step, the AEM was dried in a vacuum desiccator until no visible moisture remained. The dried membrane was then subjected to further characterization.

Supplementary Fig. 27 | N 1s fine scan and FTIR comparison of pristine PiperION and post-reaction PiperION.

To further elucidate the changes occurring in PiperION during CO electrolysis, we conducted a detailed analysis of the membrane before and after the stability test (200 mA cm⁻², 120 h) using high-resolution X-ray photoelectron spectroscopy (XPS) and Fourier-transform infrared spectroscopy (FTIR). XPS surface analysis revealed a significant decrease in N content on the surface of PiperION. Notably, after the electrolysis, N could no longer be detected on the membrane surface, indicating substantial degradation or loss of nitrogen-containing functional groups.

The FTIR spectrum of the PiperION membrane after CO electrolysis. It revealed three key changes: (1) a significant decrease in the absorption intensity around 1450–1500 cm^{-1} . It corresponds to the bending vibrations of C–N and the wagging of N–CH₃ groups in quaternary ammonium structures, such as those found in piperidinium-based ionomers.⁴ The diminished intensity in this region indicates potential degradation or de-functionalization of the quaternary ammonium groups, a phenomenon previously observed in polymeric materials bearing similar cationic moieties; (2) the disappearance of a distinct peak near $\sim 1330 \text{ cm}^{-1}$. The loss of the $\sim 1330 \text{ cm}^{-1}$ peak—attributed to the symmetric bending vibration of methyl groups in N⁺(CH₃)₂—provides further evidence of quaternary ammonium site decomposition.⁵ and (3) the emergence of a new peak near $\sim 1700 \text{ cm}^{-1}$. The enhanced peak near $\sim 1700 \text{ cm}^{-1}$ is characteristic of carbonyl (C=O) stretching vibrations, suggesting oxidative degradation or the formation of aldehyde/ketone groups. This carbonyl band is widely recognized as a marker of oxidative deterioration in polymers.⁶ These functional group changes may result from interactions between organic electrolysis intermediates (e.g., carboxylic acids, alcohols, aldehydes) and the membrane polymer, further supporting the hypothesis that PiperION membranes are susceptible to structural degradation under CO electrolysis conditions.

Specific Comment R1-4: *The authors report a decline in C₂₊ selectivity over 700 hours in the 100 cm² cell. This shift is not correlated with any catalyst metrics. Without a time-resolved analysis of the local environment or cathode state (e.g., Cu speciation), it is difficult to decouple catalyst deactivation from diaphragm-driven effects.*

Similarly, the failure of both cells at 80 °C is attributed to cathode degradation, yet supporting evidence is limited. What specific catalyst degradation pathways occur? Do these pathways differ between AEM and diaphragm systems due to membrane-electrode interactions?

Response: We thank the reviewer for this insightful comment. We agree that the observed decline in C₂₊ selectivity over time could arise from changes in either the catalyst or the local reaction environment. To partially evaluate potential catalyst evolution, especially during the initial stage of operation, we conducted cyclic voltammetry measurements (Supplementary Fig. 46). The results show that characteristic Cu facet peaks ((100), (110), and (111)) gradually diminished after 24 hours of electrolysis, indicating significant surface reconstruction of the Cu catalyst under reaction conditions. These findings suggest that the decline in product selectivity could be associated with catalyst restructuring. We have included these explanations in the revised manuscript (page 12 and Supplementary page 48).

To further elucidate the cause of cell failure at 80 °C, we conducted a 24-hour electrolysis experiment in 1 M KOH at a current density of 200 mA cm⁻². In both systems, the Faradaic efficiency for H₂ increased to approximately 50% after 24 hours of operation. However, upon replacing the cathode—while keeping the anode and diaphragm/AEM unchanged—the cell performance was fully restored to its initial state (Supplementary Fig. 34). This strongly indicates that cathode degradation is the primary cause of cell failure under elevated temperature conditions.

Electrolyte samples from the anode side and the downstream trap solution on the cathode side after 24h test were collected to trace the fate of copper (Supplementary Fig. 35). We found that at 80 °C, both the Zirfon-based cell (with Cu leaching rates of 0.114 $\mu\text{g h}^{-1} \text{ cm}^{-2}$ towards cathode and 0.606 $\mu\text{g h}^{-1} \text{ cm}^{-2}$ towards anode) and the PiperION-based cell (0.103 $\mu\text{g h}^{-1} \text{ cm}^{-2}$ towards cathode and 0.707 $\mu\text{g h}^{-1} \text{ cm}^{-2}$ towards anode) exhibited significantly higher Cu leaching rates compared to operation at room temperature. This indicates that elevated temperature accelerates the transformation of Cu into soluble species, such as [Cu(OH)₄]²⁻, which likely contributes to its rapid deactivation.

Furthermore, SEM imaging revealed substantial morphological changes in Cu particles on the cathode surfaces of both the Zirfon- and PiperION-based cells (Supplementary Fig. 36), indicating severe surface reconstruction. These observations suggest that cathode catalyst degradation is a major factor contributing to electrode failure under high-temperature conditions. Therefore, cathode degradation is likely the primary cause of cell failure in both systems operated at 80°C.

We have included these explanations in the revised manuscript (page 10 and Supplementary page 36-38).

Page 12:

After 700 hours of continuous operation, while the decline in product selectivity could be associated with Cu catalyst restructuring (Supplementary Fig. 46), the Zirfon 500+ based cell experienced gas crossover issues that ultimately led to system failure.

Supplementary Page 48:

Supplementary Fig. 46 | CV test of the Cu catalyst for CO electroreduction reaction for 10 minutes and 24h. All Cu facet-related features, including (100), (110), and (111), were disappeared, indicating that the Cu catalyst is not stable under CO electroreduction reaction. The CVs were measured under Ar conditions at a scan rate of 20 mV s^{-1} .

The experiment was conducted in a 1 M KOH with a 2.7 mL min^{-1} anolyte flow using a zero-gap electrolyzer as described previously. These experiments were performed using a BioLogic SP-300 potentiostat in a three-electrode configuration, with a polyethersulfone (PES) separator replacing a conventional membrane. A Hg/HgO reference electrode (Koslow Scientific, 5088 series, standard 1 molar solution) was inserted in the anolyte as part of the three-electrode configuration.

Page 10:

At 80°C , both systems exhibited instability within 24 hours of operation (Supplementary Fig. 28), mainly attributable to degradation of the cathode (Supplementary Fig. 32-38).

Supplementary Page 36-38:

Supplementary Fig. 34 | Faradaic efficiency of ethylene and hydrogen measured using (a) Zirfon 500+ and (b) PiperION at a fixed current density of 200 mA cm^{-2} . Restart the cell with a new cathode at 25 h. To further elucidate the cause of cell failure at 80°C , we conducted a 24-hour electrolysis experiment in 1 M KOH at a current density of 200 mA cm^{-2} . In both systems, the Faradaic efficiency for H_2 increased to approximately 50% after 24 hours of operation. However, upon replacing the

cathode—while keeping the anode and diaphragm/AEM unchanged—the cell performance was fully restored to its initial state. This strongly indicates that cathode degradation is the primary cause of cell failure under elevated temperature conditions.

Supplementary Fig. 35 | Cu leaching rates at 80 °C and room temperature for both Zirfon based cell and PiperION based cell. Electrolyte samples from the anode side and the downstream trap solution on the cathode side after 24h test were collected to trace the fate of copper. We found that at 80 °C, both the Zirfon-based cell (with Cu leaching rates of $0.114 \mu\text{g h}^{-1} \text{cm}^{-2}$ towards cathode and $0.606 \mu\text{g h}^{-1} \text{cm}^{-2}$ towards anode) and the PiperION-based cell ($0.103 \mu\text{g h}^{-1} \text{cm}^{-2}$ towards cathode and $0.707 \mu\text{g h}^{-1} \text{cm}^{-2}$ towards anode) exhibited significantly higher Cu dissolution rates compared to operation at room temperature. This indicates that elevated temperature accelerates the transformation of Cu into soluble species, such as $[\text{Cu}(\text{OH})_4]^{2-}$, which likely contributes to its rapid deactivation.

Supplementary Fig. 36 | Cathode before reaction (a and d), cathode after COR in 80°C for PiperION based cell (b and e) and Zirfon 500+ system (c and f), respectively. SEM imaging revealed substantial morphological changes in Cu particles on the cathode surfaces of both the Zirfon- and PiperION-based cells, indicating severe surface reconstruction. These observations suggest that cathode catalyst degradation is a major factor contributing to electrode failure under high-temperature conditions. Therefore, cathode degradation is likely the primary cause of cell failure in both systems operated at 80°C.

Specific Comment R1-5: While increased KOH concentration is linked to higher acetate selectivity via ketene stabilization, its possible acceleration of Zirfon degradation is not discussed. Given that ZrO_2 solubility can increase in highly alkaline environments, especially under cathodic polarization, further control experiments at varied pH would help clarify trade-offs between performance and durability.

Response: We thank the reviewer for this thoughtful comment. While it is true that elevated KOH concentrations can enhance acetate selectivity—likely through ketene intermediate stabilization—we also recognize the need to evaluate the potential durability trade-offs under such conditions, particularly with respect to membrane or diaphragm stability.

To address this concern, we performed control experiments in both 1 M and 6 M KOH using Zirfon diaphragms and conducted 24-hour electrolysis at 200 mA cm^{-2} . ICP-MS analysis of electrolyte samples from the cathode effluent and anode compartments revealed only a modest increase in Zr dissolution in 6 M KOH compared to 1 M KOH ($0.003 \rightarrow 0.002 \mu\text{g h}^{-1} \text{ cm}^{-2}$ at the cathode; $0.007 \rightarrow 0.020 \mu\text{g h}^{-1} \text{ cm}^{-2}$ at the anode). Importantly, the absolute Zr leaching rates remained very low, especially in comparison to Cu dissolution, which increased by nearly an order of magnitude under the same conditions (Supplementary Fig. 41). Furthermore, given that Zirfon 500+ contains $\sim 42.5 \text{ mg Zr cm}^{-2}$, this small degree of leaching is unlikely to impact diaphragm stability within the tested timeframes. Besides, we note that Zirfon diaphragms have been widely used in commercial alkaline water electrolyzers operating in 6 M KOH for over 10000 hours without significant stability issues, suggesting that ZrO_2 dissolution is not a major concern under such conditions.

We therefore conclude that while increased KOH concentration does slightly accelerate ZrO_2 dissolution, the primary degradation concern under high alkalinity lies with Cu instability rather than Zirfon degradation.

We have included these explanations in the revised manuscript (page 10 and Supplementary page 43).

Page 10:

However, we acknowledge that this performance gain comes at the expense of catalyst stability. Our dissolution studies reveal that higher KOH concentrations substantially accelerate Cu degradation (Supplementary Fig. 41), likely due to the enhanced formation of soluble $[\text{Cu}(\text{OH})_4]^{2-}$ species under strongly alkaline conditions.⁵¹ Therefore, although acetate selectivity is enhanced at high pH, the accompanying increase in Cu dissolution ultimately undermines long-term system stability. This trade-off underscores the importance of carefully balancing electrolyte composition to optimize both performance and durability.

Supplementary Page 43:

Supplementary Fig. 41 | Cu (a) and Zr (b) leaching rates under 1 M and 6 M KOH condition for both Zirfon based cell and PiperION based cell. We performed control experiments in both 1 M and 6 M KOH using Zirfon diaphragms and conducted 24-hour electrolysis at 200 mA cm^{-2} . ICP-MS analysis of electrolyte samples from the cathode effluent and anode compartments revealed only a modest increase in Zr dissolution in 6 M KOH compared to 1 M KOH ($0.003 \rightarrow 0.002 \mu\text{g h}^{-1} \text{ cm}^{-2}$ at the cathode; $0.007 \rightarrow 0.020 \mu\text{g h}^{-1} \text{ cm}^{-2}$ at the anode). Importantly, the absolute Zr leaching rates remained very low, especially in comparison to Cu dissolution, which increased by nearly an order of magnitude under the same conditions. Furthermore, given that Zirfon 500+ contains $\sim 42.5 \text{ mg}_{\text{Zr}} \text{ cm}^{-2}$, this small degree of leaching is unlikely to impact diaphragm stability within the tested timeframes under 6 M anolyte condition. Besides, we note that Zirfon diaphragms have been widely used in commercial alkaline water electrolyzers operating in 6 M KOH for over 10000 hours without significant

stability issues, suggesting that ZrO_2 dissolution is not a major concern under such conditions. We therefore conclude that while increased KOH concentration does slightly accelerate ZrO_2 dissolution, the primary degradation concern under high alkalinity lies with Cu instability rather than Zirfon degradation.

General Comments R2: *This manuscript reports a diaphragm-based carbon monoxide (CO) electrolyzer for multicarbon production, which replaces current unstable, high-cost AEMs with low-cost, durable Zirfon-based diaphragm. Authors were able to successfully operate 5-cm² electrolyzers at 200 mA/cm² for over 700 hours, whereas current AEM-based electrolyzers were only able to operate around ~150 hours, demonstrating highly enhanced stability. The system was further scaled up to 100-cm² cells and were also operated at 200 mA/cm² for over 700 hours, demonstrating a total acetate yield of 16.5 kg, demonstrating its commercial potential. In overall, the manuscript is well-written, and replacing the AEMs which has been considered as the key limiting factor for most alkaline-environment operating electrochemical cells with robust, cheap materials is of importance. Furthermore, the demonstration of a practical large-scale device is considered as a major breakthrough in the field. Therefore, the reviewer thinks this manuscript should be published with minor revisions.*

Response: We sincerely thank the reviewer for the positive feedback and support. We are glad the significance of our long-term stability and scale-up results was recognized. We have addressed the minor revision suggestions as detailed below and further improved the manuscript.

Specific Comment R2-1: *Porous diaphragms can allow the direct anolyte crossover to the cathode side, since they possess large, abundant pores. This can result in a few problems:*

(a) The purity of CO-reduction product can be effected. Authors should conduct K⁺ IC and pH analysis of the cathode outlet product for both diaphragm and membrane-based systems and compare their product purity.

(b) Liquid crossover can lead to the flooding of the cathode chamber, even though gas is constantly supplied. Given that the diaphragms are relatively undamaged after long-term stability tests; could this be the reason to the final degradation of cells after 700+ hours?

Response: We thank the reviewer for the thoughtful and important comment regarding potential issues related to anolyte crossover in porous diaphragm-based systems.

(a) Product purity and K⁺ crossover analysis:

To investigate differences in cathodic product purity between the Zirfon-based and PiperION-based cells, we performed 24-hour electrolysis experiments at a current density of 200 mA cm⁻² in both systems. The downstream trap solution on the cathode side was collected and analyzed to assess the purity of the products. Initially, the trap contained 3 mL of Milli-Q water with a pH of 6.9. After 24 hours of operation, the pH of the trap increased to 12.62 for the Zirfon-based cell and to 12.31 for the PiperION-based cell. Under the influence of the electric field and concentration gradients, K⁺ ions migrate toward the cathode and are subsequently carried into the trap solution by the gas stream, thereby affecting product purity. We therefore quantified the K⁺ crossover rates in both systems. The K⁺ crossover rate in the Zirfon-based cell (0.299 mg h⁻¹ cm⁻²) was found to be slightly higher than that in the PiperION-based cell (0.269 mg h⁻¹ cm⁻²), which can be attributed to the porous nature of the diaphragm in contrast to the denser structure of the AEM. Nevertheless, the difference between the two was relatively small, further confirming that Zirfon offers sufficiently low anolyte crossover and can effectively maintain compartmental separation in alkaline CO electrolysis.

(b) Potential contribution of liquid crossover to long-term degradation:

We appreciate the reviewer's insightful hypothesis regarding the role of liquid crossover in long-term degradation. We agree that porous diaphragms can, in principle, permit gradual anolyte crossover to the cathode side, which may contribute to cathode flooding and ultimately to cell failure. In our system, although we maintained a cathode backpressure of approximately 17 psi to suppress bulk liquid intrusion, we cannot completely rule out the possibility of slow liquid permeation over extended operation.

We have included these explanations in the revised manuscript (page 8 and Supplementary page 23).

Page 8:

In addition, no significant difference in product purity was observed between the two systems (Supplementary Fig. 21).

Supplementary Page 23:

Supplementary Fig. 21 | Comparison of K⁺ crossover rate and cathodic trap pH between Zirfon- and PiperION-based cells. To investigate differences in cathodic product purity between the Zirfon-based and PiperION-based cells, we performed 24-hour electrolysis experiments at a current density of 200 mA cm⁻² in both systems. The downstream trap solution on the cathode side was collected and analyzed to assess the purity of the products. Initially, the trap contained 3 mL of Milli-Q water with a pH of 6.9. After 24 hours of operation, the pH of the trap increased to 12.62 for the Zirfon-based cell and to 12.31 for the PiperION-based cell. Under the influence of the electric field and concentration gradients, K⁺ ions migrate toward the cathode and are subsequently carried into the trap solution by the gas stream, thereby affecting product purity. We therefore quantified the K⁺ crossover rates in both systems. The K⁺ crossover rate in the Zirfon-based cell (0.299 mg h⁻¹ cm⁻²) was found to be slightly higher than that in the PiperION-based cell (0.269 mg h⁻¹ cm⁻²), which can be attributed to the porous nature of the diaphragm in contrast to the denser structure of the AEM. Nevertheless, the difference between the two was relatively small, further confirming that Zirfon offers sufficiently low anolyte crossover and can effectively maintain compartmental separation in alkaline CO electrolysis.

Specific Comment R2-2: Does utilizing diaphragms with higher porosities enable more CO to crossover to the anode side and negatively impact the single-pass conversion efficiency of the system? It would be better if the authors could detect the content of CO at the anolyte outlet and calculate the SPCEs for both diaphragm and membrane-based systems.

Response: We appreciate the reviewer's insightful comment. In our experiments, none of the diaphragms tested exhibited noticeable CO crossover at the early stage of operation. However, we agree that in theory, higher diaphragm porosity leads to a lower bubble point. Once the bubble point drops below the applied back pressure of 17 psi (used in our setup), CO crossover to the anode becomes significant. At that point, due to the risk of CO mixing with O₂ at the anode side, the diaphragm is no longer suitable for safe or meaningful testing.

In our current study, we did not detect any CO at the anode outlet during the initial operation (Supplementary Fig. 9b), confirming that gas crossover was negligible under the chosen conditions.

Specific Comment R2-3: In Fig.4(c), the cell potentials for 3M and 6M conditions are lower than 1M condition at lower current densities, but is reversed at high current densities. Considering that higher ionic conductivity can be provided with concentrated electrolytes, what could be the reason for this?

Response: We thank the reviewer for this thoughtful question. While increasing KOH concentration generally reduces cell voltage at low current densities by improving ionic conductivity, the opposite trend is observed at high current densities due to mass transport limitations and interfacial effects.

As demonstrated in Ren et al. (doi: 10.1016/j.joule.2023.08.008), the cell voltage initially decreases with increasing KOH concentration at 100 mA cm⁻². However, at 500 mA cm⁻², further increases in KOH concentration lead to higher cell voltages. This is attributed to the elevated viscosity and reduced CO/water transport near the catalyst surface, resulting in greater concentration overpotentials. These observations are consistent with our own experimental findings.

We have now included this explanation and supporting reference in the revised manuscript (page 10).

Page 10:

Fig. 4 | Electrochemical properties of Zirfon 500+ at elevated temperatures. a | Temperature-dependent tests of Zirfon 500+ among different current density. b | Stability performance of Zirfon 500+ at 60°C with a fixed current density of 200 mA cm⁻². c | Faradaic efficiency for all detectable products and corresponding cell voltages of Zirfon 500+ based cell measured with different KOH concentrations (1 M, 3 M, and 6 M). **While increasing KOH concentration generally reduces cell voltage at low current densities by improving ionic conductivity, the opposite trend is observed at high current densities due to mass transport limitations and interfacial effects.**⁴⁵ The cell was operated at a fixed current density of 200 mA cm⁻² at 60°C, a 40-60 nm Cu nanoparticle cathode, a NiFeO_x/Ni foam anode, with CO fed at a rate of 50 sccm.

Specific Comment R2-4: *Is there a specific reason that the authors applied this diaphragm-based systems to CO electrolysis? Would it be possible to demonstrate a CO₂ reduction electrolyzer with this system?*

Response: We thank the reviewer for the thoughtful comment. We chose the CO reduction system in this study with the intention of ultimately integrating it with upstream CO production via solid oxide electrolysis cells (SOECs) to construct a tandem CO₂-to-multicarbon platform. SOECs are currently more mature for high-efficiency CO generation, and techno-economic analyses have shown that this route can lead to significantly lower CO production costs compared to direct CO₂ electrolysis (doi: 10.1038/s44359-025-00045-1).

Additionally, recent studies have begun to explore diaphragm integration in CO₂ electrolysis systems (doi: 10.1002/adsu.202500285), further supporting the practical potential of diaphragms in both CO and CO₂ electroreduction applications.

Specific Comment R2-5: *In overall, there are some formatting errors in Page 12, 17 in manuscript and Page 28, 34 of supporting document.*

Response: We thank the reviewer for pointing out the formatting issues. We have carefully reviewed and corrected the formatting errors on Page 12 and 17 of the main manuscript, as well as on Page 28 and 34 of the Supporting Information. The revised versions have been uploaded accordingly. Including:

- Corrected a spelling error in the word “favricated” → “fabricated”.
- All chemical formulae have been reformatted to include proper subscript and superscript styling (e.g., “ZrO2” → “ZrO₂”, “NiFeOx” → “NiFeO_x”).
- The notation for weight percent has been standardized throughout the manuscript, replacing “wt.%” with the consistent and journal-recommended format “wt%”.

General Comments R3: *This manuscript systematically investigated a series of diaphragms as alternative separators for alkaline CO electrolysis and compared them with alkaline polyelectrolyte membranes (AEMs). The CO electrolysis performance using some of these diaphragms was comparable to that of state-of-the-art AEMs at current densities ranging from 50 to 400 mA cm⁻². It is noted that many reports on alkaline CO electrolysis have achieved high current densities up to several A cm⁻². So, a question arises: why did this paper conduct performance evaluations at relatively lower current densities? Admittedly, the performance of the 100 cm² Zirfon-based cell was truly striking. While the paper presented numerous performance comparisons, it lacked an analysis of the reasons for performance changes. For example, could some in-situ spectroscopy methods be used to study the reaction mechanism at the cathode interface, to clarify the influence mechanism of the diaphragm on CORR? The specific comments are as follows:*

Response: We thank the reviewer for the thoughtful and constructive comments.

Regarding the current density range selected for this study, we acknowledge that some state-of-the-art CO electrolysis systems have reported current densities exceeding 1 A cm^{-2} . It is important to note that such performance is often achieved only under transient or short-duration conditions, which do not reflect long-term operational stability. Furthermore, as demonstrated in alkaline water electrolyzers, diaphragms are prone to gas crossover at high current densities due to increased transmembrane flux, which may compromise Faradaic efficiency and safety. In our work, we prioritized realistic continuous operation. The Zirfon-based systems were evaluated at up to 400 mA cm^{-2} over extended periods, which we believe provides a more representative and rigorous assessment of diaphragm suitability for durable CO electrolysis.

We agree that mechanistic insights into the effect of the diaphragm on CO reduction are highly valuable. Due to certain objective limitations, we were not able to incorporate in-situ mechanistic studies into this work. As this study primarily focuses on electrochemical performance and scalability, we fully recognize the importance of further investigating catalyst–electrolyte–separator interactions. In future work, we plan to integrate in-situ spectroscopic techniques—such as in-situ X-ray characterization—to probe changes in the local reaction environment and catalyst surface states under different diaphragm configurations.

Specific Comment R3-1: *Using XPS survey spectra to analyze the degradation of PiperION is not a reasonable approach. What is the degradation pathway of piperidine N? It is recommended to conduct a specific analysis on N1s spectra.*

Response: We thank the reviewer for the insightful comment. We agree that relying solely on XPS survey spectra to assess chemical degradation is not sufficient. In response, we have revised our analysis to include a more specific discussion of the N 1s spectra (Supplementary Fig. S27a), and incorporated relevant insights into the potential degradation pathway of piperidinium nitrogen under CO electrolysis conditions.

Our high-resolution N 1s spectra indicate the complete loss of the quaternary ammonium nitrogen signal after electrolysis, consistent with the degradation or de-functionalization of piperidinium cations. While XPS is a surface-sensitive technique and susceptible to artifacts, this observation is further supported by FTIR analysis, which reveals:

1. A significant decrease in peaks assigned to C–N bending and N–CH₃ wagging ($1450\text{--}1500 \text{ cm}^{-1}$),
2. Disappearance of the $\sim 1330 \text{ cm}^{-1}$ signal from methyl groups on N⁺(CH₃)₂ units, and
3. Emergence of a carbonyl C=O band ($\sim 1700 \text{ cm}^{-1}$), suggesting oxidative decomposition of the polymer backbone.

These spectral changes collectively support a chemical degradation mechanism involving both quaternary ammonium group loss and backbone oxidation.

Piperidinium-based cations are known to degrade under alkaline electrochemical conditions through mechanisms such as:

- Nucleophilic attack by OH[−], leading to ring opening;
- Hofmann elimination, causing fragmentation of the cationic structure;
- Oxidative degradation initiated by electrolysis-derived reactive intermediates (e.g., aldehydes, acids, or radicals).

Such degradation pathways are well documented for polymeric quaternary ammonium groups and provide a mechanistic basis for the nitrogen loss observed in our study.

We have incorporated these discussions in the revised manuscript (pages 8, 22 and Supplementary Information pages 28–29), as suggested.

Page 8:

Additionally, XPS and FTIR revealed the disappearance of the N 1s peak initially present in the pristine PiperION membrane (Supplementary Fig. 26 and 27), suggesting the degradation of PiperION during COR.

Page 22:

FTIR spectra were obtained using a Thermo Scientific Nicolet iS20 FTIR spectrometer with an attenuated total reflection (ATR) accessory. The spectra were recorded with 0.125 cm^{-1} intervals at room temperature. Each spectrum was averaged over 32 scans. All samples were rinsed thoroughly with Milli-Q water, vacuum-dried, and analyzed directly without further treatment.

Supplementary Page 28-29:

XPS experimental sample pretreat: After completion of the electrolysis, the AEM was carefully removed from the electrochemical cell and immediately rinsed with a large volume of Milli-Q water to remove any residual electrolyte from the membrane surface. Following the rinsing step, the AEM was dried in a vacuum desiccator until no visible moisture remained. The dried membrane was then subjected to further characterization.

Supplementary Fig. 27 | N 1s fine scan and FTIR comparison of pristine PiperION and post-reaction PiperION.

To further elucidate the changes occurring in PiperION during CO electrolysis, we conducted a detailed analysis of the membrane before and after the stability test (200 mA cm^{-2} , 120 h) using high-resolution X-ray photoelectron spectroscopy (XPS) and Fourier-transform infrared spectroscopy (FTIR). XPS surface analysis revealed a significant decrease in N content on the surface of PiperION. Notably, after the electrolysis, N could no longer be detected on the membrane surface, indicating substantial degradation or loss of nitrogen-containing functional groups.

The FTIR spectrum of the PiperION membrane after CO electrolysis. It revealed three key changes: (1) a significant decrease in the absorption intensity around $1450\text{--}1500\text{ cm}^{-1}$. It corresponds to the bending vibrations of C–N and the wagging of N–CH₃ groups in quaternary ammonium structures, such as those found in piperidinium-based ionomers.⁴ The diminished intensity in this region indicates potential degradation or de-functionalization of the quaternary ammonium groups, a phenomenon previously observed in polymeric materials bearing similar cationic moieties; (2) the disappearance of a distinct peak near $\sim 1330\text{ cm}^{-1}$. The loss of the $\sim 1330\text{ cm}^{-1}$ peak—attributed to the symmetric bending vibration of methyl groups in N⁺(CH₃)₂—provides further evidence of quaternary ammonium site decomposition.⁵ and (3) the emergence of a new peak near $\sim 1700\text{ cm}^{-1}$. The enhanced peak near $\sim 1700\text{ cm}^{-1}$ is characteristic of carbonyl (C=O) stretching vibrations, suggesting oxidative degradation or the formation of aldehyde/ketone groups. This carbonyl band is widely recognized as a marker of oxidative deterioration in polymers.⁶ These functional group changes may result from interactions between organic electrolysis intermediates (e.g., carboxylic acids, alcohols, aldehydes) and the membrane polymer, further supporting the hypothesis that PiperION membranes are susceptible to structural degradation under CO electrolysis conditions.

Specific Comment R3-2: Regarding the stability tests of Fig. 4 (Zirfon) and the ones using the PiperION membrane in the SI, what are the reasons for the rapid increase in hydrogen evolution in the later stage and the rapid decrease in ethylene in the early stage? I do not think that the rapid performance decline at the initial 20 hours is due to the defects of membrane materials.

Response: We thank the reviewer for the insightful comment. We agree that the rapid performance decline in the early stage is not due to membrane degradation. This is supported by the fact that reused membranes could still deliver comparable performance to pristine ones in subsequent tests (Supplementary Fig. 22).

The sharp increase in hydrogen evolution in the later stage is more likely related to interfacial issues between the cathode and the PiperION membrane. Specifically, we suspect that the higher interfacial resistance (Supplementary Fig. 18) in this system may lead to uneven electric field distribution across the electrode surface, accelerating localized deactivation.

During the initial ~20 hours of electrolysis, we observe a sharp decline in ethylene selectivity and a corresponding increase in acetate production across all stability tests, including systems using both the Zirfon diaphragm and the PiperION membrane. We attribute this trend to surface reconstruction of the Cu catalyst (Supplementary Fig. 46), which alters the local reaction environment and product pathway. As the catalyst surface evolves during this early stage, the selectivity gradually shifts, with acetate emerging as the dominant C₂ product in the stabilized phase.

We have included these explanations in the revised manuscript (Supplementary Page 18, 24, 48).

Supplementary Page 18:

Supplementary Fig. 18 | Comparison of catalyst-membrane/separator interface. a | Scheme of catalyst-membrane/separator interface contact condition. **b |** Thickness measurement of separator and membrane before compression and after reaction. **c |** Cathode ohmic overpotential and corresponding cell voltages for different current density.

Supplementary Page 24:

Supplementary Fig. 22 | Faradaic efficiency of acetate, ethylene and hydrogen and corresponding cell voltages measured using PiperION at a fixed current density of 200 mA cm⁻². The 5 cm² zero-gap CO electrolyzer was operated at room temperature with a 1 M KOH electrolyte at 3 mL min⁻¹, a 40–60 nm Cu nanoparticle cathode, a NiFeO_x/Ni foam anode, with CO fed at a rate of 50 sccm.

The sharp increase in hydrogen evolution in the later stage is not likely related to the membrane degradation. This is supported by the fact that reused membranes could still deliver comparable performance to pristine ones in subsequent. Specifically, we suspect this is caused by the interfacial issues between the cathode and the PiperION membrane (Supplementary Fig. 18). The higher interfacial resistance in this system may lead to uneven electric field distribution across the electrode surface, accelerating localized deactivation.

Supplementary Page 48:

Supplementary Fig. 46 | CV test of the Cu catalyst for CO electroreduction reaction for 10 minutes and 24h. All Cu facet-related features, including (100), (110), and (111), were disappeared, indicating that the Cu catalyst is not stable under CO electroreduction reaction. The CVs were measured under Ar conditions at a scan rate of 20 mV s⁻¹. The experiment was conducted in a 1 M KOH with a 2.7 mL min⁻¹ anolyte flow using a zero-gap electrolyzer as described previously. These experiments were performed using a BioLogic SP-300 potentiostat in a three-electrode configuration, with a polyethersulfone (PES) separator replacing a conventional membrane. A Hg/HgO reference electrode (Koslow Scientific, 5088 series, standard 1 molar solution) was inserted in the anolyte as part of the three-electrode configuration.

Specific Comment R3-3: *Why does the cell voltage increase at high current densities as the alkali concentration increases?*

Response: We thank the reviewer for this thoughtful question. While increasing KOH concentration generally reduces cell voltage at low current densities by improving ionic conductivity, the opposite trend is observed at high current densities due to mass transport limitations and interfacial effects.

As demonstrated in Ren et al. (doi: 10.1016/j.joule.2023.08.008), the cell voltage initially decreases with increasing KOH concentration at 100 mA cm⁻². However, at 500 mA cm⁻², further increases in KOH concentration lead to higher cell voltages. This is attributed to the elevated viscosity and reduced CO/water transport near the catalyst surface, resulting in greater concentration overpotentials. These observations are consistent with our own experimental findings.

We have now included this explanation and supporting reference in the revised manuscript (page 10).

Page 10:

Fig. 4 | Electrochemical properties of Zirfon 500+ at elevated temperatures. a | Temperature-dependent tests of Zirfon 500+ among different current density. b | Stability performance of Zirfon 500+ at 60°C with a fixed current density of 200 mA cm⁻². c | Faradaic efficiency for all detectable products and corresponding cell voltages of Zirfon 500+ based cell measured with different KOH concentrations (1 M, 3 M, and 6 M). While increasing KOH concentration generally reduces cell voltage at low current densities by improving ionic conductivity, the opposite trend is observed at high current densities due to mass transport limitations and interfacial effects.⁴⁵ The cell was operated at a fixed current density of 200 mA cm⁻² at 60°C, a 40-60 nm Cu nanoparticle cathode, a NiFeO_x/Ni foam anode, with CO fed at a rate of 50 sccm.

Specific Comment R3-4: *What causes the sudden increase in cell voltage at 100 h in Fig. 5? In several stability testing plots, the selectivity for acetate was low at the beginning, and the selectivity for alcohol products was not given. Is there a possibility that alcohol products are oxidized to acetate at the anode side?*

Response: We thank the reviewer for this insightful observation. The sudden increase in cell voltage was due to a drop in electrolyte pH, caused by the accumulation of acetate during continuous operation. Specifically, the pH of the circulating electrolyte decreased from ~14 to ~8, leading to a noticeable rise in cell voltage—an effect consistent with previous literature reports (doi: 10.1038/s41929-022-00828-w). Upon replacing the electrolyte, the pH increased and the cell voltage returned to normal. The subsequent voltage fluctuations observed during long-term testing were similarly associated with periodic electrolyte replacement. We added the notification in the figure. We have now added a corresponding note to the figure caption to clarify this. (Page 11)

As for the Faradaic efficiency of alcohols, we measured ~3–5% FE in freshly collected samples. However, alcohols are easily oxidized in alkaline media over time (doi: 10.1038/s41929-022-00828-w). Since the product analysis was conducted using the recirculated electrolyte, the alcohol content was largely transferred into acetate.

Page 11:

Fig. 5 | Performance of 100 cm² CO Electrolysis Cell. a | Photograph of 100 cm² CO electrolyzer. b | Faradaic efficiency of acetate, ethylene and hydrogen and corresponding cell voltages measured using Zirfon 500+ at a fixed current density of 200 mA cm⁻². The 100 cm² zero-gap CO electrolyzer was operated with a 1 M KOH electrolyte at a flow rate of 60 mL min⁻¹, a 40–60 nm Cu nanoparticle cathode, a NiFeO_x/Ni foam anode, with CO fed at a rate of 400 sccm and at room temperature. **The sudden rise in cell voltage was caused by a pH drop in the electrolyte due to acetate buildup during extended electrolysis. Replacing the electrolyte restored the pH and lowered the cell voltage accordingly.** c | Comparison of acetate production rate and operational stability among state-of-the-art CO electrolyzers. Dashed lines represent the total acetate yield achieved during the whole reported operation period.^{5,8,50-56} See Supporting Excel files for detailed source data.